# A comprehensive sulfate and DOM framework to assess methylmercury formation and risk in subtropical wetlands

Brett A. Poulin [1] ✉, Michael T. Tate[2], Sarah E. Janssen [2], George R. Aiken[3,4] & David P. Krabbenhoft [2]

Wetlands play a vital role in contaminant cycling and uptake. Understanding how sulfate ($SO_4^{2-}$) influences the conversion of inorganic mercury (Hg(II)) to toxic methylmercury (MeHg) is critical for predicting wetland responses to land use and climate change. Here, we sampled surface and pore waters across $SO_4^{2-}$ gradients in three freshwater Everglades wetlands to assess linkages between $SO_4^{2-}$, MeHg, dissolved organic matter (DOM), and inorganic sulfide (S(–II)). Increasing $SO_4^{2-}$ concentrations increase S(–II) and DOM concentrations and DOM aromaticity. MeHg concentration show a unimodal response to surface water $SO_4^{2-}$, which reflect high Hg(II) methylation at low-to-intermediate $SO_4^{2-}$ concentration (2-12 mg/L) and low Hg(II) methylation at higher $SO_4^{2-}$ concentrations (> 12 mg/L). MeHg concentrations in surface waters correlate positively with MeHg concentrations in prey fish. The coherent biogeochemical relationships between $SO_4^{2-}$ and MeHg concentrations and biologic uptake improve MeHg risk assessment for aquatic food webs and are globally relevant due to anthropogenic and climate-driven increases in $SO_4^{2-}$.

Mercury (Hg) is a global pollutant that, when converted to neurotoxic methylmercury (MeHg), poses severe risks to wildlife and humans[1]. Freshwater wetlands are important locations for the cycling of Hg between environmental compartments, with both source (e.g., atmospheric deposition)[2] and sink processes (e.g., photo-reduction of inorganic divalent Hg (Hg(II)) to elemental Hg (Hg(0)))[3,4] governing Hg(II) concentrations in wetland waters. The microbial conversion of Hg(II) to MeHg is a critical step in the Hg cycle responsive to hydrologic and biogeochemical perturbations due to internal (e.g., nutrient cycling, land and water management) and external factors (e.g., climate change variables)[5], which can have cascading effects that influence MeHg risk to biota[6]. MeHg risk is defined as the potential for MeHg to be formed within or proximal to aquatic systems that results in efficient MeHg assimilation in the aquatic food web. For over three decades[7,8], sulfate ($SO_4^{2-}$), a terminal electron acceptor for anaerobic microbial metabolism, has been identified as an important geochemical predictor of the potential of MeHg formation in freshwater wetlands[9–15]. However, due to the individual biogeochemical complexities of sulfur (S), organic carbon (C), and Hg in the environment, there is still uncertainty in the response between $SO_4^{2-}$ concentrations and MeHg formation in wetlands and food web uptake[16–18], the locations where MeHg formation occurs (e.g., sediments[19] versus water column versus periphyton)[20,21], and the direct or indirect role of $SO_4^{2-}$ reducing bacteria in the conversion of Hg(II) to MeHg[19,22]. A revisitation of linkages between $SO_4^{2-}$ concentrations and the potential for MeHg formation and uptake in biota is needed. Although wetland MeHg concentrations reflect the balance between Hg(II) methylation[3,4,19] and MeHg demethylation[9,23], differences in MeHg concentration within and across boreal[15] and subtropical wetlands[9,24] are primarily attributed to differences in Hg(II) methylation. Freshwater environments globally are threatened by increasing $SO_4^{2-}$ concentrations due to expansion in agricultural sulfur (S) use (primarily as

[1]Department of Environmental Toxicology, University of California Davis, Davis, CA, USA. [2]U.S. Geological Survey, Mercury Research Laboratory, Madison, WI, USA. [3]U.S. Geological Survey, Water Mission Area, Boulder, CO, USA. [4]Deceased: George R. Aiken. ✉e-mail: bapoulin@ucdavis.edu

elemental S[25,26], climate-driven increases in mineral weathering that release $SO_4^{2-}$ in boreal and Arctic systems[27,28], and sea-level rise that delivers marine $SO_4^{2-}$ to coastal wetlands[29].

Sulfate ($SO_4^{2-}$) exerts primary and secondary controls in MeHg formation through a combination of redox-dependent geochemical and microbial processes. From a geochemical perspective, in freshwater wetlands with low $SO_4^{2-}$ (<1 mg/L), Hg(II) aqueous speciation is primarily governed by the binding of Hg(II) to thiol groups in dissolved organic matter (DOM)[30]. Hg(II) speciation can be drastically altered in wetlands with appreciable $SO_4^{2-}$ and labile organic carbon, as dissimilatory $SO_4^{2-}$ reduction produces inorganic sulfide (S(–II)), which outcompetes DOM thiol groups for Hg(II), resulting in the formation of nano-particulate metacinnabar (nano-β-HgS)[31]. Further, microbially mediated dissimilatory $SO_4^{2-}$ reduction, which utilizes soil organic matter (SOM), can catalyze both the release of highly aromatic DOM from soil and sediment[29,32,33] and the abiotic sulfurization of DOM, enriching the DOM in thiol groups[34]. Nascent nano-β-HgS particles are smaller and less crystalline when formed under environmental conditions (i.e., low Hg(II) to DOM)[31,35] and in the presence of DOM of higher aromaticity[36]. From a microbial perspective, $SO_4^{2-}$ can have primary and secondary effects on microorganisms that convert Hg(II) to MeHg, which have the requisite *hgcAB* gene pair[37]. In pure culture, $SO_4^{2-}$-reducing bacteria are highly efficient at converting Hg(II) to MeHg[38–40], with the most pronounced MeHg formation by heterotrophic bacteria existing in cultures with intermediate S(–II) concentrations[39], high DOM aromaticity[40], and high DOM thiol content[41]. However, in nature, *hgcAB*+ organisms span diverse phylogenetic groups beyond $SO_4^{2-}$-reducing bacteria, including methanogenic archaea and fermentative organisms[19,42]. Notably, in two systems with environmental[19] and manipulated[22] gradients in $SO_4^{2-}$, the diversity and abundance of the *hgcAB* gene pair decreased with increasing $SO_4^{2-}$. The combination of $SO_4^{2-}$ effects on Hg(II) bioavailability for methylation and activity of the microbial community are understood to contribute to the nonlinear response observed between $SO_4^{2-}$ and MeHg[8,9,11–15,43]. Given the potential for fluctuations in $SO_4^{2-}$ concentrations in wetlands in response to land management (e.g., agricultural practices), water management (e.g., storm water treatment, phosphate mitigation via alum additions)[44,45], or climate change processes[27–29], as well as the uncertainties in the geochemical and microbial responses pertaining to MeHg formation, ecosystem-scale studies are needed to solidify linkages between $SO_4^{2-}$, MeHg formation, and subsequent MeHg uptake in aquatic food webs.

Here, field campaigns were conducted along hydrologic transects in three differing wetlands of the freshwater Florida Everglades, a managed wetland ecosystem with high atmospheric Hg deposition rates[2,46]. The wetland transects exhibited distinct horizontal geochemical gradients in $SO_4^{2-}$ spanning from low $SO_4^{2-}$ (<0.5 mg/L) wetlands that receive $SO_4^{2-}$ solely from the atmosphere to those highly enriched in $SO_4^{2-}$ (>65 mg/L) from agricultural runoff[45,47]. Across geochemical $SO_4^{2-}$ gradients, surface waters and pore waters were sampled for pertinent biogeochemical parameters, including filtered and particulate Hg(II) and MeHg (f.Hg(II), p.Hg(II), f.MeHg, p.MeHg), S species ($SO_4^{2-}$, S(–II)), and dissolved organic carbon (DOC) concentration and DOM composition (specific ultraviolet absorbance at 254 nm ($SUVA_{254}$)). We present data showing coherence between $SO_4^{2-}$ concentrations in wetlands, shifts in DOM composition, and MeHg formation across a comprehensive field study, which included the examination of MeHg concentrations across vertical gradients in wetlands (between surface and pore waters) and lateral gradients across wetlands. Lastly, relationships between MeHg concentrations in surface waters were compared to MeHg concentrations in prey fish (*Gambusia holbrooki*), an indicator species with a short life span (≤6 months) that reflects recent MeHg risk to the aquatic food web[17,43,48]. The findings advance a new ecosystem-scale conceptual model of the effects of $SO_4^{2-}$ on Hg risk to the aquatic food web and are discussed in context of parallel efforts that quantified the effects of $SO_4^{2-}$ on DOM thiol content[34], the abundance and metabolic capabilities of *hgcAB*+ microorganisms[19], downgradient impacts on MeHg formation and biouptake in Everglades National Park[48], and strategies to mitigate the risk of Hg to wildlife and humans. This information will be vital to anticipate future effects of $SO_4^{2-}$ on Hg(II) methylation potential across subtropical wetlands in response to management actions, climate change, and international efforts to mitigate atmospheric Hg releases in the environment.

## Results and discussion

### Chemistry of canals contributing to freshwater Everglades wetlands

Canals, which drain Lake Okeechobee and the Everglades agricultural area[45], exert considerable control on the biogeochemistry of waters in the three water conservation areas studied (water conservation areas 2A and 3A (WCA-2A and WCA-3A), and the Arthur R. Marshall Loxahatchee National Wildlife Refuge (LOX)), and exhibit distinct chemistries (Tables S1–S3 and Fig. 1). The levee-6 (L-6) canal, which discharges into northwest WCA-2A, drains water with DOM of higher aromaticity (DOM $SUVA_{254}$ = 3.48 ± 0.16 L/mg m; avg ± 1 standard deviation) and higher concentrations of DOC (33.1 ± 4.2 mg/L), $SO_4^{2-}$ (64.7 ± 32.4 mg/L), and chloride (Cl⁻) (131 ± 37.4 mg/L) compared to the levee-28 (L-28) canal (DOM $SUVA_{254}$ = 3.30 ± 0.34 L/mg m; DOC = 19.6 ± 2.9 mg/L; $SO_4^{2-}$ = 6.84 ± 3.65 mg/L; Cl⁻ = 32.8 ± 18.0 mg/L), the latter draining into northwest WCA-3A (Fig. 1). The Palm Beach Canal, which is an outer canal along the northeast perimeter of LOX, contains water of lower DOM aromaticity (DOM $SUVA_{254}$ = 2.63 ± 0.34 L/mg m) and intermediate DOC, $SO_4^{2-}$, and Cl⁻ concentration compared to the L-6 and L-28 canals (DOC = 19.9 ± 12.5 mg/L; $SO_4^{2-}$ = 43.9 ± 40.6 mg/L; Cl⁻ = 96.2 ± 58.7 mg/L). Of the four Hg fractions measured from the three canals over the study (f.Hg(II), f.MeHg, p.Hg(II), p.MeHg; $n$ = 23), Hg was primarily present as f.Hg(II) across all sites (Table S3). Particulate fractions accounted for <25% of the total Hg in canal waters (Fig. S1). Of the filtered fractions (f.Hg(II) and f.MeHg), f.MeHg was only responsible for 9-11% and f.Hg(II) accounted for the remaining 89–91% of the THg. Further, concentrations of f.Hg(II) and f.MeHg were similar across the three canals, with mean values ranging from 0.67–0.74 ng/L and 0.07–0.10 ng/L, respectively. One outlier was observed of elevated f.MeHg (1.11 ng/L) from the L-28 canal (08/18/2014). Differences in the chemistry of canal waters are interpreted to reflect upgradient agricultural practices that enrich waters in inorganic anions ($SO_4^{2-}$, Cl⁻) from agricultural practices[45,47] and mobilize highly aromatic DOM from peat soils[33], which has important implications on the biogeochemical cycling of Hg in downgradient wetlands.

### Surface and pore water biogeochemistry across wetland transects

The chemistry of surface and pore waters across the three wetlands exhibits distinct lateral and vertical differences that reflect proximal canal inputs and biogeochemical transformations of organic C, S, and Hg that arise from these inputs. The highest spatial resolution sampling was conducted in August 2014 (Fig. 2) across WCA-2A ($n$ = 16) and WCA-3A ($n$ = 15), and December of 2015 in LOX ($n$ = 5), with sampling events in other years conducted at a reduced number of sites or only including surface waters (Fig. S2 and Table S2)[49]. Surface water $SO_4^{2-}$ concentrations of wetlands were comparable to adjacent canals at the top of each transect (i.e., near km = 0) and exhibited a concentration hierarchy of WCA-2A > LOX > WCA-3A (Fig. 2a–c). With increased distance from canals, surface water $SO_4^{2-}$ concentrations consistently decreased (e.g., in Fig. 2, $SO_4^{2-}$ decreased from 65.8 to 50.6 mg/L, 22.5 to <0.5 mg/L, and 4.1 to <0.5 mg/L across transects in WCA-2A, LOX, and WCA-3A, respectively). Further, surface waters were sub-oxic to anoxic in all three wetlands, immediately downgradient of canal inputs (Fig. S3a–c), with dissolved

oxygen concentrations generally increasing with distance from canal inputs[49]. The oxidation-reduction potential (ORP) of wetlands surface water and pore waters (Fig. S3d–f) reflected the degree of canal influence, with significantly lower ORP in pore waters of WCA-2A (–271 ± 12.3 mV; average ± std) compared to WCA-3A (–168.7 ± 18.4 mV) or the interior of LOX (LOX8, –25 mV) (Welch's $t$-test; $p < 0.001$). At the most downgradient site in WCA-3A, which is proximal to the L-29 canal, notable increases in $SO_4^{2-}$ and $Cl^-$ concentrations were observed (Fig. 2b) due to canal backflow into WCA-3A, as supported by the Everglades depth estimation network (EDEN) flow model (Fig. 1)[50]. The Florida Everglades is naturally a low $SO_4^{2-}$ environment, with concentrations between 0.1–1.0 mg/L in regions unimpacted by agricultural S inputs[45], and therefore $SO_4^{2-}$ concentrations observed in interior WCA wetlands greatly exceeded background concentrations (up to ~70-fold) and were distributed across the entire range observed (≤0.5–72.0 mg/L; Fig. S4), creating vastly different water quality conditions across downgradient freshwater wetlands.

Several lines of evidence support the prominence of dissimilatory $SO_4^{2-}$ reduction in wetlands downgradient of canal inputs, explaining the decrease in surface water $SO_4^{2-}$ concentration with increased distance from canals. First, in sediment pore waters, $SO_4^{2-}$ concentrations were significantly lower than corresponding surface waters (Mann–Whitney rank sum test, $p < 0.001$), by ≤60.5, ≤26.5, and ≤3.5 mg/L in WCA-2A, LOX, and WCA-3A, respectively (Fig. 2a–c). Second, concentrations of inorganic sulfide (S(–II)), a byproduct of dissimilatory $SO_4^{2-}$ reduction, were highest in pore waters of WCA-2A (S(–II) = 1.6–9.4 mg/L), intermediate at locations in LOX and WCA-3A nearest canals (S(–II) ≤0.45 and ≤0.16 mg/L, respectively), and below detection limit at locations in LOX and WCA-3A with $SO_4^{2-}$ ≤ 0.5 mg/L (S(–II) ≤0.01 mg/L) (Fig. 2d–f); this hierarchy matched the magnitude of negative ORP across the wetlands (Fig. S3d–f). Although pore water $SO_4^{2-}$ and S(–II) concentrations varied with distance from the canal (Fig. 2d–f), particularly across WCA-2A, a significant positive correlation was observed between pore water $SO_4^{2-}$ and S(–II) concentrations across all sites (Fig. S5; $R^2 = 0.64$, $p < 0.001$; $n = 60$). Notably, S(–II) concentrations were above the detection limit (S(–II) >0.01 mg/L) in anoxic surface waters of WCA-2A (Fig. S6) but below the detection limit in surface waters from WCA-3A and LOX. Third, in wetland locations with evidence of $SO_4^{2-}$ reduction, we observed consistent decreases in the molar ratio of $SO_4^{2-}$ to $Cl^-$ with increased distance from canals ($SO_4^{2-}/Cl^-$; Fig. 2g–i), the latter being a conservative analyte inert to biogeochemical transformations. Concentrations of $Cl^-$ were largely uniform across the WCA-2A and WCA-3A (Fig. S7)[49] and therefore decreases in the molar $SO_4^{2-}/Cl^-$ are interpreted to be primarily due to

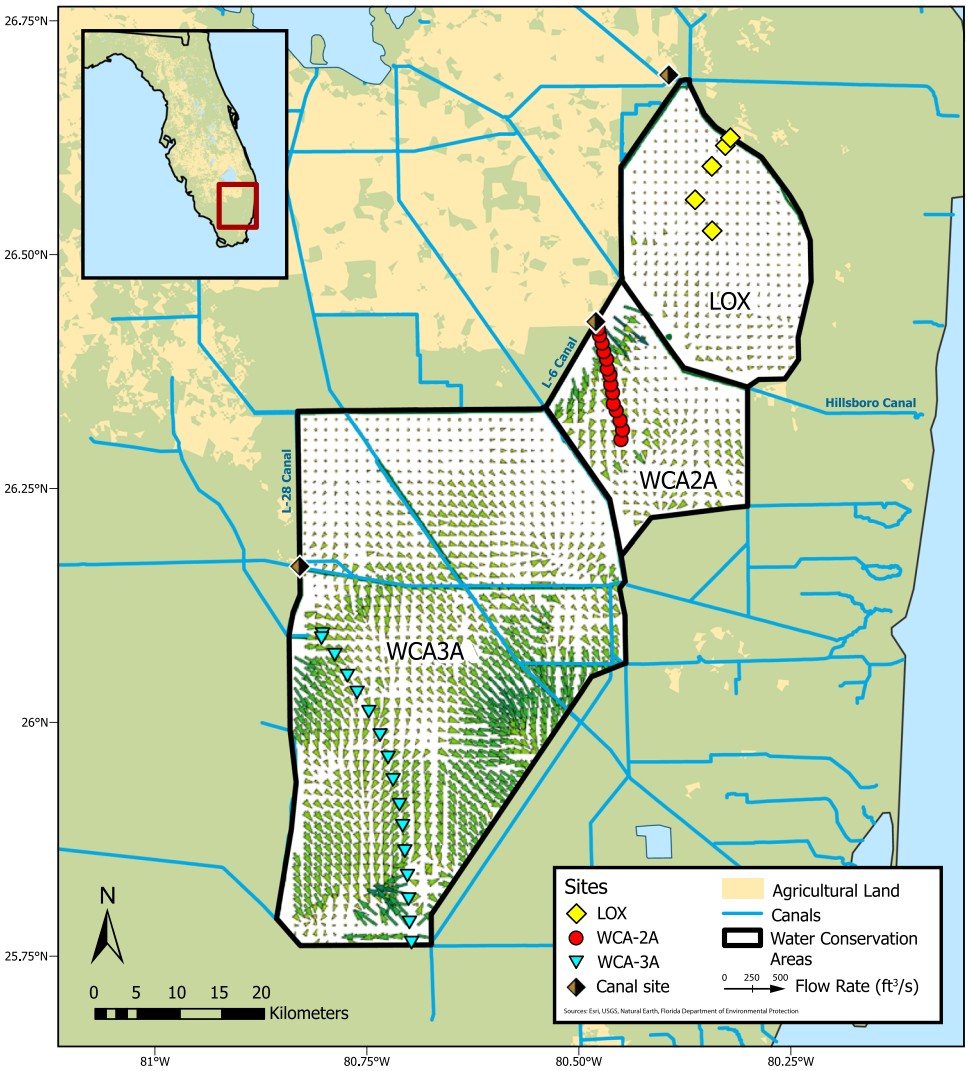

**Fig. 1 | Map of the Florida Everglades.** Locations of surface water and pore water collection across hydrologic flow paths in water conservation areas 2A and 3A, and a geochemical gradient in the Arthur R. Marshall Loxahatchee National Wildlife Refuge (LOX). Flow vectors are presented based on the Everglades Depth Estimation Network (EDEN) model (vectors shown for 11/17/2015)[50].

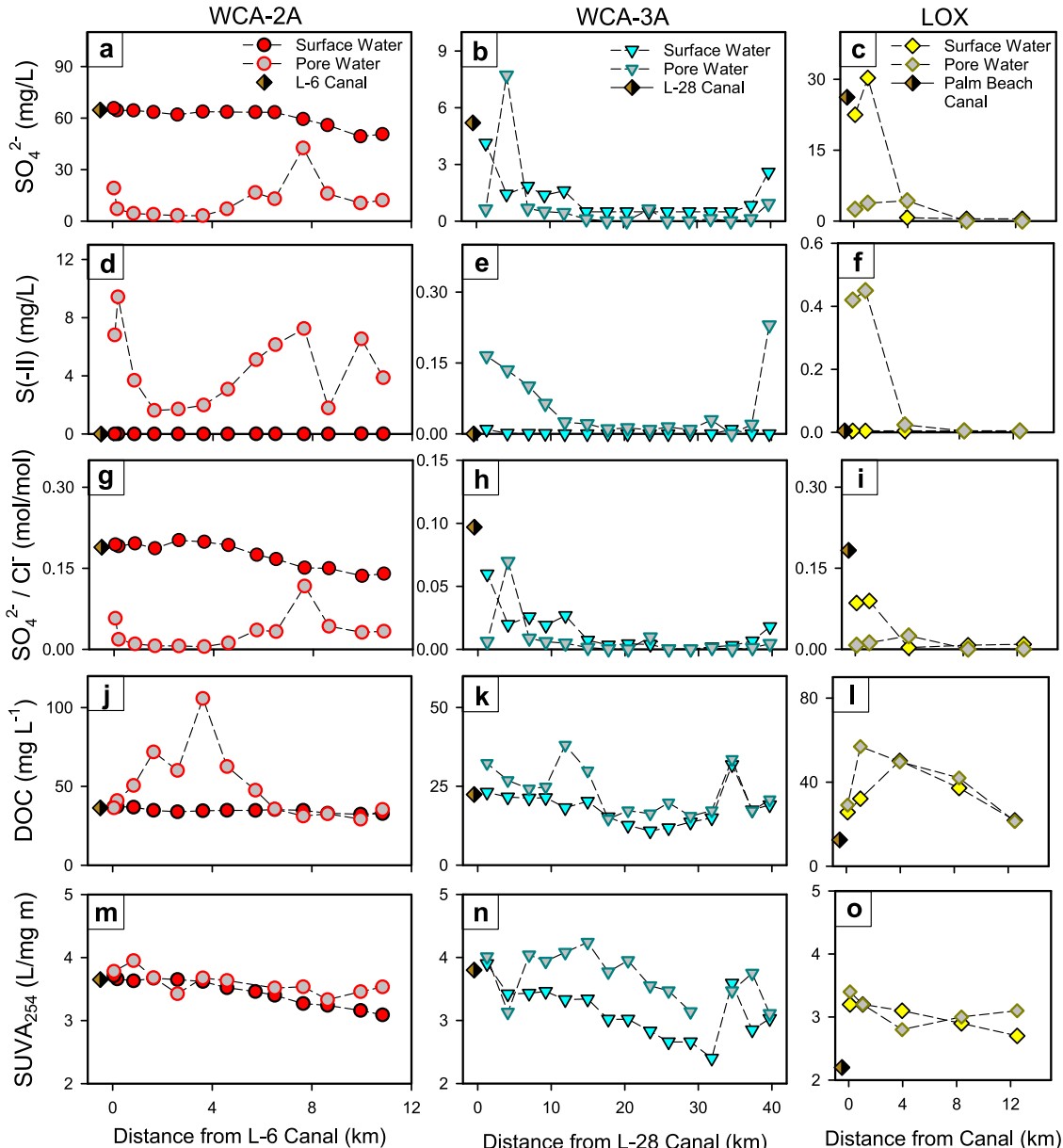

**Fig. 2 | Lateral and vertical sulfate (SO$_4^{2-}$) and dissolved organic matter (DOM) trends across wetlands.** Biogeochemical constituents of (**a**–**c**) SO$_4^{2-}$ concentration, **d**–**f** inorganic sulfide (S(-II)) concentration, **g**–**i** the molar ratio of SO$_4^{2-}$ to chloride (SO$_4^{2-}$ / Cl$^-$), **j**–**l** dissolved organic carbon (DOC) concentration, and (**m**–**o**) DOM specific ultraviolet absorbance at 254 nm (SUVA$_{254}$) in water conservation area 2A (WCA-2A) (August, 2014), WCA-3A (August, 2014), and the Arthur R. Marshall Loxahatchee National Wildlife Refuge (LOX) as a function of distance from canal inputs (November, 2015). For each transect, canal conditions are presented at the time of wetland sampling. Dashed lines are presented to guide the eye. Oxidation-reduction potential (ORP) and dissolved oxygen (O$_2$) data were presented in Fig. S3.

microbial SO$_4^{2-}$ reduction, consistent with previous observed increases in δ$^{34}$S of SO$_4^{2-}$ due to reduction of isotopically light S[47], the abundance of genes of SO$_4^{2-}$ reducing bacteria (e.g., *dsrA*)[19], and landscape SO$_4^{2-}$ models[51]. These wetland transects document SO$_4^{2-}$ contamination of the WCAs, extending >10 km downgradient of active points of canal water release (WCA-2A, WCA-3A) or through intrusion from canals (LOX). Wetland SO$_4^{2-}$ concentrations governed the spatial extent of SO$_4^{2-}$ reducing conditions, with S(-II) accumulating in pore waters across all sites as a function of SO$_4^{2-}$ availability (Fig. S5) and extending to surface waters in WCA-2A (Fig. S6).

Canal contributions also had a marked influence on the DOC concentration and DOM composition within wetlands. Across wetland gradients, surface water DOC concentrations and DOM SUVA$_{254}$, a proxy for aromatic C content[52], were consistent between neighboring canals and adjacent wetland sites (Fig. 2j–o). In wetland pore waters, however, DOC concentrations were significantly higher compared to surface waters (Mann–Whitney rank sum test, $p < 0.001$, $n = 70$), particularly in regions with active SO$_4^{2-}$ reduction near canal inputs (Fig. 2j–l), and DOM in pore waters was of modestly higher aromaticity (DOM SUVA$_{254}$; Welch's *t*-test, $p < 0.001$, $n = 70$; Fig. 2m–o). Higher DOC concentration and greater DOM aromaticity in wetland pore waters near canal inputs compared to surface waters are interpreted to be the result of SO$_4^{2-}$-stimulating degradation of peat by SO$_4^{2-}$ reducing bacteria[33] that release DOM of higher aromaticity (e.g., phenolic groups)[29,32]. Further, the anoxic (Fig. S3a–c), sulfidic conditions that prevail due to SO$_4^{2-}$ inputs are known to prevent enzymatic degradation of aromatic DOM[53] and promote the sulfurization of DOM, the latter enriching DOM in thiol functional groups proportional to SO$_4^{2-}$ inputs

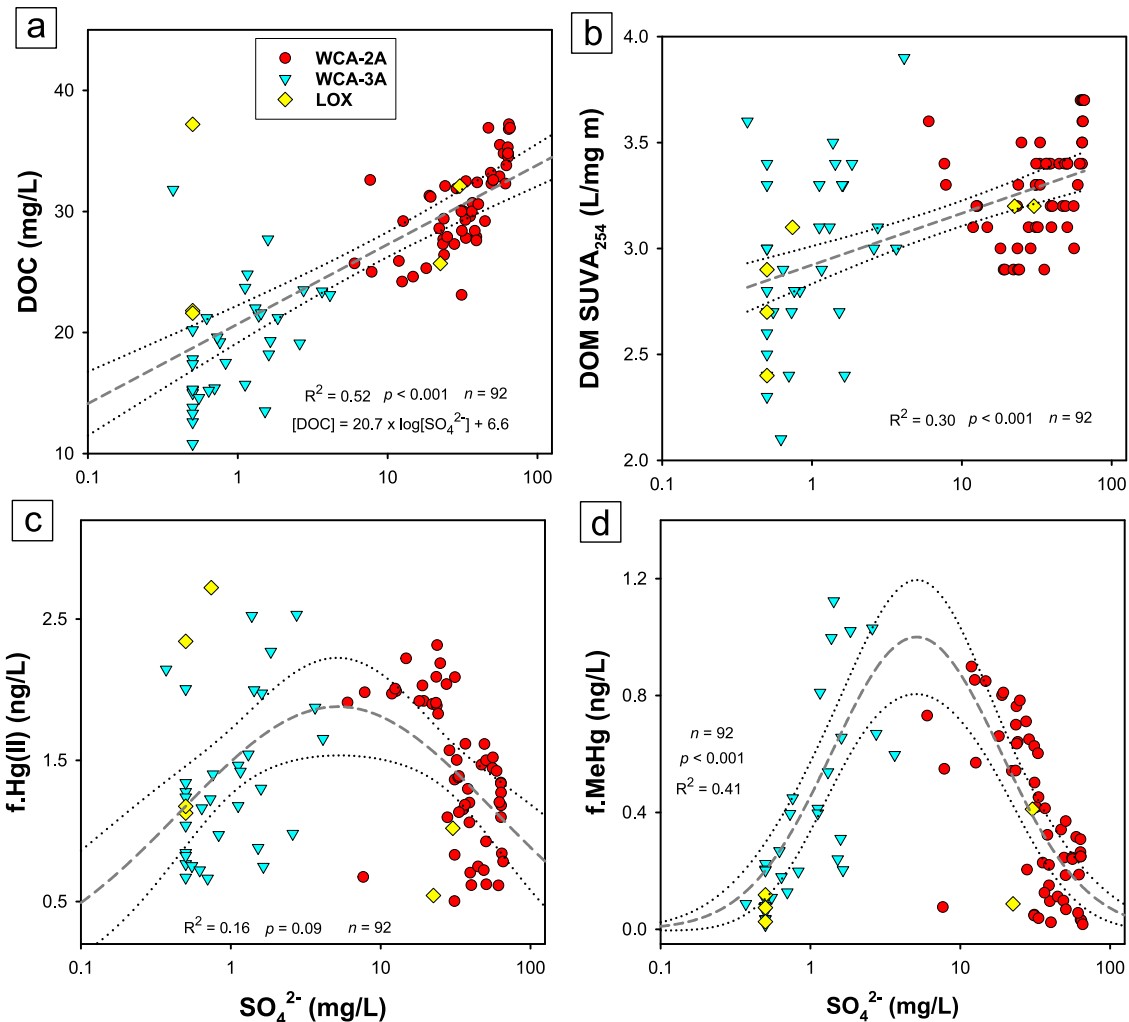

**Fig. 3 | Dissolved organic matter (DOM) and mercury responses to sulfate ($SO_4^{2-}$).** Semi-log scatter plots between the concentration of sulfate and (**a**) dissolved organic carbon (DOC) concentration, (**b**) DOM specific ultraviolet absorbance at 254 nm ($SUVA_{254}$), (**c**) filtered inorganic divalent Hg (f.Hg(II)) concentration, and (**d**) filtered methylmercury (f.MeHg) concentration. In subplots a and b, dashed gray lines and dotted blank lines present the linear regression and 95% confidence intervals between $SO_4^{2-}$ and DOC concentration and DOM $SUVA_{254}$. In subplots c and d, dashed gray lines and dotted black lines present the unimodal equations and 95% confidence intervals of the fit, respectively (Eq. 1, S1; Fig. S9).

to wetlands[34]. With increased distance from canals in WCA-2A and WCA-3A, surface water DOC concentrations decreased (e.g., from 37.0 to 32.7 and 23.1 to 15.0 mg/L, respectively) and DOM $SUVA_{254}$ decreased (e.g., from 3.7 to 3.1 and 3.9 to 2.4 L/mg m, respectively) (Fig. 2m, n), likely the result of the photochemical transformation of DOM[54] and autochthonous DOM production in wetland surface waters[55]. The decadic absorption coefficient of DOM at 254 nm ($\alpha_{254}$), which is a bulk DOM metric that scales positively with DOC concentrations and DOM $SUVA_{254}$, showed similar trends across the three wetlands with higher DOM $\alpha_{254}$ near canal inputs and in wetlands with higher $SO_4^{2-}$[49].

The effects of $SO_4^{2-}$ on DOM quantity and composition were observed across all wetland surface waters over the seven sampling events of this study (Fig. 3a–b), with significant positive correlations between $SO_4^{2-}$ and DOC concentration ($R^2 = 0.52$, $p < 0.001$; $n = 92$) and $SO_4^{2-}$ concentration and DOM $SUVA_{254}$ ($R^2 = 0.30$, $p < 0.001$; $n = 92$). These relationships are interpreted to arise from upgradient and within-wetland processes resulting from the biogeochemical cascade due to $SO_4^{2-}$ effects on DOM mobilization from peat[29,32,33]. Greater variability is observed between correlations of DOC concentration and DOM $SUVA_{254}$ and $SO_4^{2-}$ concentration at lower $SO_4^{2-}$ concentrations (Fig. 3a, b), which we interpret to be the result of canal waters with

intermediate $SO_4^{2-}$ concentrations stimulating aromatic DOM release and $SO_4^{2-}$ reduction that depletes available $SO_4^{2-}$ (Fig. S5). In summary, canal water $SO_4^{2-}$ inputs lower the wetland redox status and stimulate S(–II) production and the release of aromatic DOM across distinct lateral and vertical geochemical gradients within all three Everglades wetlands, and subsequently these biogeochemical changes govern the geochemical nature of Hg(II)[31,36] and bioavailability for methylation[39,40].

## Mercury speciation and biological uptake across wetland transects

The lateral and vertical spatial distribution of Hg fractions was tightly coupled to the biogeochemical cascade linked to canal $SO_4^{2-}$ inputs. Across all wetlands, particulate Hg fractions were measured (p.Hg(II), p.MeHg)[49] and were a minor fraction (accounted for ≤10% of the total Hg) (Fig. S1), consistent with previous observations[4], and thus the analyses in this study focused on f.Hg(II) and f.MeHg. Concentrations of f.Hg(II) in wetland surface waters were consistent with those of adjacent canals (Fig. 4a–c) and showed modest variability from up-to-downgradient along wetland transects and between surface and pore waters. Modestly higher f.Hg(II) concentrations were observed in pore waters with elevated DOC concentration, but f.Hg(II) concentration did

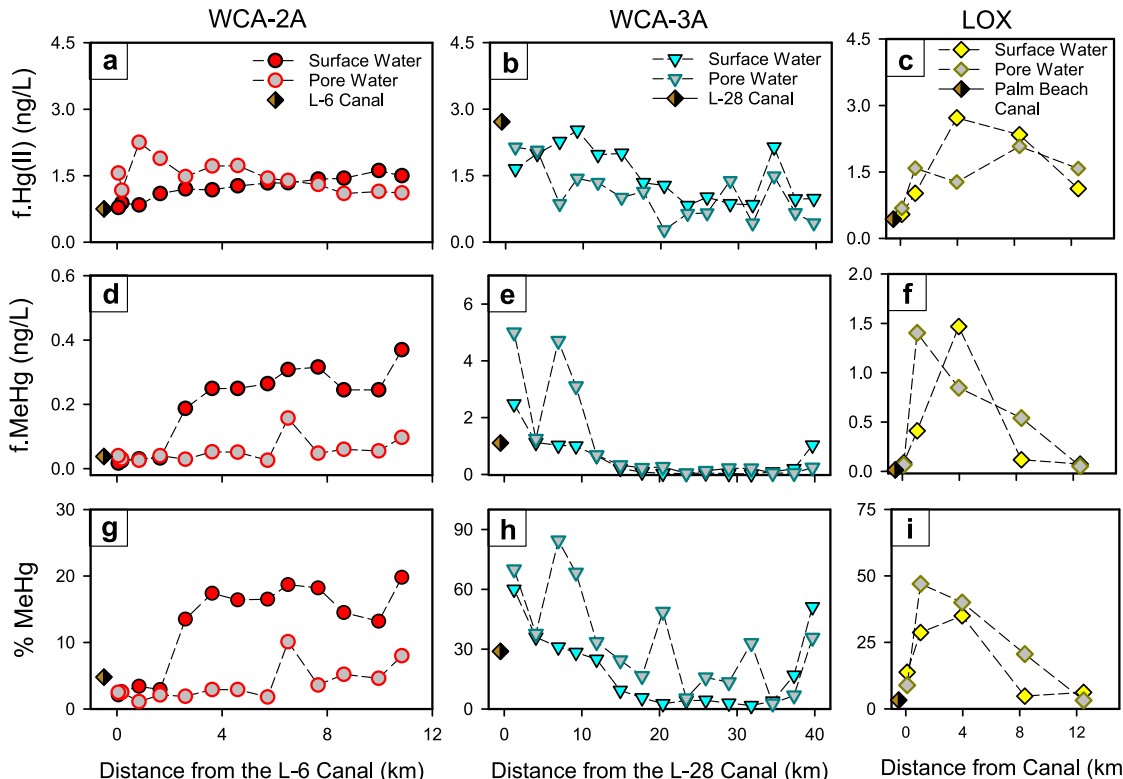

**Fig. 4 | Lateral and vertical mercury trends across wetlands.** Concentrations of filtered (**a**–**c**) inorganic Hg (f. Hg(II)), **d**–**f** methylmercury (f. MeHg), and (**g**–**i**) the percentage of total Hg as MeHg (%MeHg) in water conservation area 2A (WCA-2A), WCA-3A, and the Arthur R. Marshall Loxahatchee National Wildlife Refuge (LOX) as a function of distance from canal inputs (August 2014). For each transect, canal conditions are presented. Dashed lines are presented to guide the eye.

not linearly correlate to DOC concentration ($R^2 = 0.06$; $n = 112$), consistent with previous observations in the Everglades[4]. Rather, f.Hg(II) concentration exhibited a nonlinear relationship to $SO_4^{2-}$ concentration, where f.Hg(II) concentrations were low at $SO_4^{2-} < 1.0$ mg/L, increased between 1 and 10 mg/L $SO_4^{2-}$, and decreased at $SO_4^{2-} > 10$ mg/L (Fig. 3c). There was considerable variability in the relationship between f.Hg(II) and $SO_4^{2-}$ concentration, and thus the fit of data was not of statistical significance ($p = 0.09$; SI Section S1). We interpret the relationship to be from $SO_4^{2-}$ (1) increasing f.Hg(II) concentration at low-to-intermediate $SO_4^{2-}$ concentration, due to increases in DOC concentration and DOM $SUVA_{254}$ (Fig. 3a, b) that enhance complexation of Hg(II)[30], and (2) decreasing f.Hg(II) concentration at high $SO_4^{2-}$ concentration through S(–II) scavenging f.Hg(II) and forming nano-β-HgS(s)[31,36] that aggregate[35]. Although additional source and sink processes also influence f.Hg(II) concentrations in Everglades wetlands, including rainfall delivery of f.Hg(II)[2,46], photo-reduction of Hg(II) to Hg(0)[3,4], and partitioning of Hg(II) to peat[43], $SO_4^{2-}$ exhibits notable control on the concentration of f.Hg(II), which can limit MeHg formation in wetlands.

The concentration of f.MeHg and percentage of total Hg as MeHg (% MeHg) showed dramatic differences between the three wetlands, with distinct vertical trends between surface and pore waters and lateral trends across wetlands. The %MeHg term is used to normalize f.MeHg concentration data to the total Hg concentration at a given location, as f.Hg(II) concentrations varied by an order of magnitude within and between the studied wetlands (e.g., Fig. 4a–c and S8a, b) and is the precursor for f.MeHg[9,56]. Canals draining into WCAs typically had f.MeHg concentrations and %MeHg of 0.10 ng/L and 10%, respectively (Table S3 and Fig. S8), with rare instances of higher concentrations (e.g., L-28 canal during August 2014; Fig. 4e)[49]. In WCA-2A, which had the highest $SO_4^{2-}$ loading, f.MeHg concentrations and %MeHg were low in both surface and pore waters at the first 4 wetland locations downgradient of the L-6

canal input (0.01–0.04 ng/L and 1.1–3.4 %MeHg, respectively; Fig. 4d and Fig. S8). Further downgradient, average f.MeHg concentrations and %MeHg were higher in surface waters (0.27 ± 0.05 ng/L and 16.4 ± 2.3 %MeHg, respectively; ±1 standard deviation) compared to pore waters (0.06 ± 0.04 ng/L and 4.5 ± 2.8 %MeHg, respectively) or the L-6 canal (Mann–Whitney rank sum test, $p < 0.001$, $n = 14$). In WCA-3A, which had lower $SO_4^{2-}$ loading than WCA-2A, f.MeHg concentrations and %MeHg were highest near the L-28 canal input, were greater in pore waters compared to surface waters, and decreased monotonically with distance from the canal (Fig. 4e, h and Fig. S8d, f) consistent with decreases in surface water $SO_4^{2-}$ (Fig. 2b). The highest f.MeHg and %MeHg levels observed across all three wetlands were in pore waters (≤5.0 ng/L and ≤84.5%, respectively) and surface waters (≤2.5 ng/L and ≤60%, respectively) of WCA-3A near the L-28 canal input, which is consistent with other studies on MeHg spatial distribution across the system[4,43,51]. In LOX, which had the widest range in $SO_4^{2-}$ concentrations, f.MeHg and %MeHg were low adjacent to the canal (≤0.09 ng/L and ≤13.8%, respectively) (Fig. 4f, i), increased to a maximum approximately 4 km from the canal (≤1.50 ng/L and ≤47%, respectively), and declined in the interior of the wetland (≤0.07 ng/L and ≤6.2%) were $SO_4^{2-}$ concentrations were <1 mg/L; similar behavior was observed in both surface and pore waters f.MeHg concentration across the LOX wetland $SO_4^{2-}$ gradient.

Across all wetland surface waters sampled over the seven field campaigns, a nonlinear unimodal trend was observed between surface water $SO_4^{2-}$ and f.MeHg concentrations (Fig. 3d). A nonlinear, 3-parameter equation was used to fit field data, defined in Eq. 1,

$$f.MeHg = a \times e^{\left(-0.5 \times \frac{\left[\frac{\ln\left(SO_4^{2-}/x_0\right)}{b}\right]^2}{SO_4^{2-}}\right)} \quad (1)$$

where $a$, $b$, and $x_0$ were 11.9, 1.30, and 27.6, respectively ($R^2 = 0.41$, $p < 0.001$; $n = 93$). A maximum in f.MeHg concentration was observed at a $SO_4^{2-}$ concentration between 4 and 6 mg/L, and lower f.MeHg concentrations were observed at lower and higher $SO_4^{2-}$ concentrations. There was good agreement between observed and predicted f.MeHg concentrations at observed f.MeHg concentrations ≥0.1 ng/L (Fig. S9a) with no systematic difference in residual f.MeHg concentrations (i.e., the difference between observed and predicted f.MeHg concentrations) across the range of $SO_4^{2-}$ concentration (Fig. S9b), indicating the unimodal fit was robust across the conditions of the three wetlands. At both high (>30 mg/L) and low $SO_4^{2-}$ concentration (<1 mg/L), which corresponded to low f.MeHg concentration (<0.1 ng/L), a modest discrepancy was noted between observed and predicted f.MeHg concentrations (Fig. S9a). Across all wetlands, the %MeHg was significantly correlated ($R^2 = 0.77$, $p < 0.001$; $n = 168$) to the concentration of f.MeHg in water (Fig. S10), which has been observed in the Everglades[9,19] and other systems[10,15] to correspond with faster measured rates of MeHg production. Thus, wetlands with higher aqueous f.MeHg and %MeHg are interpreted to reflect recent MeHg production, as MeHg that is produced is susceptible to sink processes (e.g., photo-demethylation[57] and adsorption to peat)[58].

Vertical and lateral spatial trends in f.MeHg concentration and %MeHg in response to $SO_4^{2-}$ reflect a combination of (1) the redox state of wetland surface and pore waters, (2) geochemical processes governing Hg(II) aqueous speciation[31,36] and bioavailability[39–41,59], and (3) the microbial communities with the pre-requisite $hgcAB$ gene pair for methylation[37]. Below we detail the observations from low to high $SO_4^{2-}$ concentration across the three wetlands and interpret environmental f.MeHg concentrations to reflect the balance between Hg(II) methylation[3,4,19] and MeHg demethylation[9,23], with differences primarily being driven by differences in Hg(II) methylation[9,15,24]. At $SO_4^{2-}$ concentrations <1 mg/L, observed in the middle of WCA-3A and LOX, f.MeHg concentrations were low in surface and pore waters, interpreted to be a combination of the absence of S(–II) and higher redox state of the water column and wetland sediments (Fig. S3), lower f.Hg(II) concentration, DOC concentration, and DOM SUVA254 (Fig. 3a–c), and lower DOM thiol content[34]; all of these conditions decrease the bioavailability of Hg(II) to methylation[39–41]. Although higher $hgcAB$ gene abundance was observed in peat of low $SO_4^{2-}$ Everglades wetlands[19], the low ambient MeHg observed under these conditions is attributed to high wetland redox state and low Hg(II) bioavailability. The modest disagreement between observed and predicted f.MeHg at $SO_4^{2-}$ concentrations <1 mg/L (Fig. S9a) simply reflects the low MeHg formation under low, ambient levels of $SO_4^{2-}$ that are not well fit using a 3-parameter unimodal function. At intermediate $SO_4^{2-}$ concentrations (2–6 mg/L), such as in LOX and the upgradient portions of WCA-3A (Fig. 4e, f), f.MeHg concentrations were elevated in sulfidic pore waters compared to oxygenated surface waters, which is interpreted to reflect enhanced MeHg formation in surficial wetland sediments and advection and diffusion to surface waters[3]. The highest f.MeHg concentration and %MeHg were observed under these conditions (Figs. 3d, 4e), where microbial $hgcAB$ gene abundance is elevated in sediments[19] and sulfidic pore waters have high DOC concentration and DOM of more aromatic and thiol group content[34], which promote Hg(II) methylation[39–41] likely as nano-β-HgS(s)[31,36]. At locations of high $SO_4^{2-}$ concentrations (>12 mg/L) in WCA-2A downgradient from canals (Fig. 4d, g), f.MeHg concentration and %MeHg were notably higher in surface waters compared to pore waters. This observation cannot be explained by MeHg formation in sediments and advective or diffusive flux[60,61], but may be attributed to MeHg formation by periphyton associated with emergent and submerged aquatic vegetation in the water column[20,21]. The shallow wetlands studied here are unlikely to support bulk water column methylation observed in deeper, thermally stratified waters where suspended particulate materials accumulate[62,63]. Lastly, at locations of high $SO_4^{2-}$ concentration in WCA-2A near the canal, f.MeHg concentration was low in both surface and pore waters (Fig. 4d, g), which is attributed to high S(–II) conditions in surface and pore waters that decrease the concentration and bioavailability of f.Hg(II) (Fig. 3c) via the formation[31,36] and aggregation[35] of nano-β-HgS(s) and the low microbial abundance of communities with the $hgcAB$ genes[19]. Despite the elevated concentrations of highly aromatic, sulfurized DOM[34] under these sulfidic conditions, crystalline nano-β-HgS(s)[36] will form and dramatically decrease f.Hg(II) concentrations in the wetland pore and surface waters (Figs. 3c, 4a) and contributes to decreased MeHg formation[64]. Despite many inferential studies that concluded direct links between $SO_4^{2-}$ loading, the activity of $SO_4^{2-}$ reducing bacteria, and MeHg production[3,4,9,21,45], a recent microbial metagenomic analysis of sediments across these same sites did not observe $hgcAB+$ organisms with the genes for dissimilatory $SO_4^{2-}$ reduction[19]. Rather, MeHg concentration and experimental methylation rates were governed by synergy between the $hgcAB$ abundance of other microbial clades and DOM composition (SUVA254)[19]. We interpret that $SO_4^{2-}$ inputs directly alter Hg(II) geochemistry and bioavailability and indirectly alter the microbial metabolism, which stimulates MeHg formation at low-to-intermediate $SO_4^{2-}$ concentrations (2–12 mg/L) by $SO_4^{2-}$ reducing bacteria consuming fermentation products and/or stimulating methanogenic activity through syntrophy[19].

The field data underlying these biogeochemical relationships span multiple years (2012–2019) and seasons (May and December) (Table S2) and leverage geochemical[34], hydrological, and microbial insights[19], providing a comprehensive dataset to develop a framework for MeHg production and risk. Taken together, a highly consistent trend is observed across all wetland surface waters between concentrations of $SO_4^{2-}$, the DOM $\alpha_{254}$ (which scales positively with increases in DOC concentrations and DOM SUVA254), and f.MeHg concentration (Fig. 5). We interpret the spread in f.MeHg at a given $SO_4^{2-}$ concentration (Figs. 3d, 5) to be the effects of seasonality on MeHg formation, as noted previously[4,17], with higher observed MeHg formation in the spring and summer months (Fig. S11) due to higher rates of microbial metabolism and regular rainfall delivering Hg(II)[3] to wetlands. The unimodal relationship here between $SO_4^{2-}$ and f.MeHg concentration (Fig. 3d, 5) is consistent with the conceptual model first proposed by Gilmour and Henry (1994)[8] but incorporates recent innovations in the understanding of Hg(II) nano-scale geochemistry and the complexities of microbial Hg(II) methylation, which provides mechanistic insights that explain MeHg formation across the greater Everglades[16] and in response to increases[43] and decreases in $SO_4^{2-}$ loading[51]. We assert that the clear coherence in the relationship between $SO_4^{2-}$ and f.MeHg concentration here is a product of the study design, with field campaigns that sampled lateral hydrologic flow paths from canal sources downgradient across wetlands, resolved vertical gradients between wetland surface waters and sediment pore waters, and coupled routine biogeochemical data (e.g., Figs. 2–5) with parallel efforts on $SO_4^{2-}$ effects on DOM composition[34] and abundance and metabolic capabilities of $hgcAB+$ microorganisms[19]. Although the assertion of $SO_4^{2-}$ is a master variable on MeHg formation in the Everglades is not in agreement with conclusions from state environmental reports[65,66], which may be due to differences in study design or approaches used for biogeochemical characterization, the observed findings are congruent with studies on the effects of $SO_4^{2-}$ on MeHg formation in diverse wetlands[9–15].

The concentrations of f.MeHg in wetland surface waters significantly correlated with the concentration of MeHg in gambusia (Fig. 6; $R^2 = 0.68$; $p < 0.001$; $n = 33$), a resident fish species[17]. The total

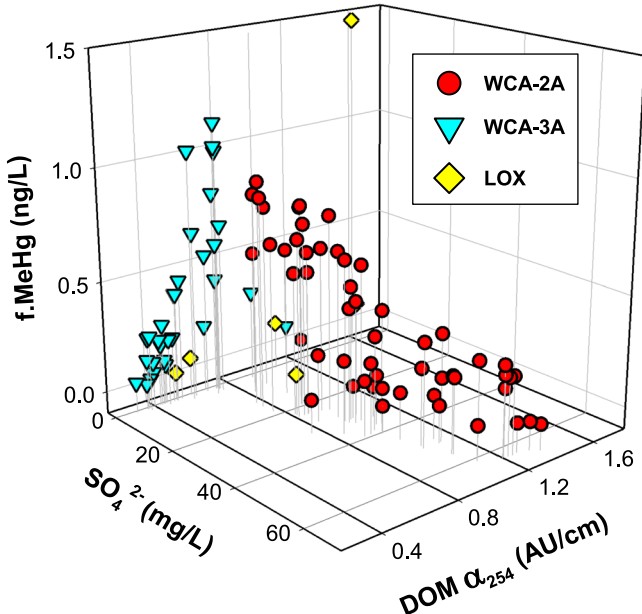

**Fig. 5 | Relationship between sulfate (SO₄²⁻), dissolved organic matter (DOM), and methylmercury (MeHg).** Three-dimensional scatter plot between the surface water concentration of SO₄²⁻, DOM absorbance at 254 nm (DOM $\alpha_{254}$), and concentration of filtered methylmercury (f.MeHg) in surface water across water conservation area 2A (WCA-2A), WCA-3A, and the Arthur R. Marshall Loxahatchee National Wildlife Refuge (LOX).

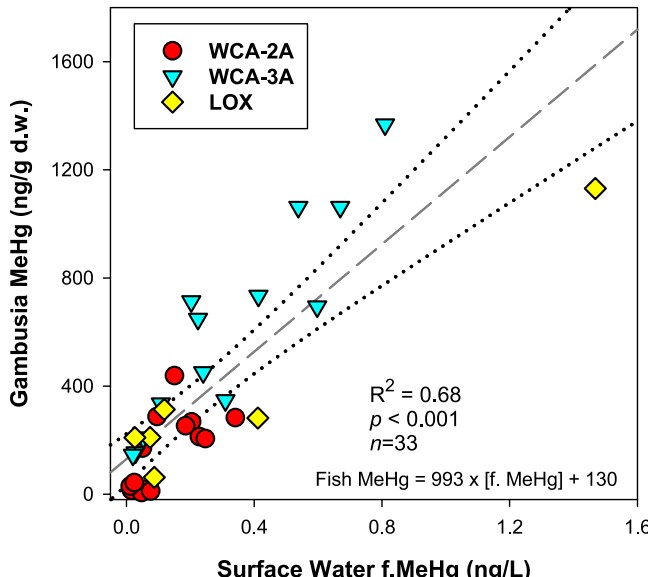

**Fig. 6 | Linkage between methylmercury (MeHg) in water and fish.** Scatter plot between concentration of filtered methylmercury (f.MeHg) in surface water and concentration of MeHg of gambusia across water conservation area 2A (WCA-2A), WCA-3A, and the Arthur R. Marshall Loxahatchee National Wildlife Refuge (LOX). The dashed gray line is the linear fit to the data, and dotted black lines correspond to the 95% confidence intervals of the fit.

Hg concentration of *Gambusia* was primarily MeHg (91.2 ± 14.1%, $n = 33$)[49]. Provided that these resident prey fish live relatively short lives (≤6 months) and feed on a combination of periphyton and zooplankton[17], the data provide strong support that SO₄²⁻ promotes MeHg formation in surficial sediments and compartments of the water column resulting in enhanced risk of MeHg bioaccumulation at the

base of the aquatic food web. Figure 7 synthesizes a framework on the association between SO₄²⁻, DOM, and the potential for MeHg formation in proximity to aquatic food webs, which dictates MeHg risk. We further conclude that water concentrations of f.MeHg may be a good proxy to predict food web risk in the Everglades due to the strong relationship observed here and within Everglades National Park[48]. The highest gambusia MeHg concentrations were observed at locations with intermediate SO₄²⁻ concentrations, with fish MeHg concentrations exceeding 1000 ng/g dry weight, similar to that reported previously across the WCAs[67] and greater Everglades ecosystem[16,68]. Taken together, this study shows coherence in the underlying hydrologic and biogeochemical processes that govern MeHg risk in the freshwater Everglades (Fig. 7), which support the use of SO₄²⁻ concentration to model risk across the freshwater ecosystem[51] and to forecast the influence of water management decisions on MeHg formation and food web bioaccumulation in Everglades National Park[48].

### Strategies to mitigate MeHg risk in wetlands

Wetlands globally are being affected by increases in SO₄²⁻ due to a variety of new pressures, which will influence Hg cycling. Agricultural use of S-containing fertilizers has increased 200% in the last 30 years in the United States[25], and climate change processes are increasing SO₄²⁻ concentrations across the globe. For example, enhanced mineral weathering in the Arctic has increased SO₄²⁻ concentrations in the Mackenzie River by 45% in the last 60 years[28], and sea-level rise is threatening coastal wetlands with marine SO₄²⁻ that will result in the contraction of freshwater ecosystems[29]. The comprehensive framework presented here (Fig. 7) for the Florida Everglades support that efforts to decrease SO₄²⁻ concentrations from agricultural sources have promise to decrease aqueous and biological MeHg concentrations in wetlands (Figs. 5, 6), offering a local strategy to mitigate Hg contamination. Local reductions in SO₄²⁻ concentration or load may yield a relatively fast response, as previous work in the Everglades demonstrated that declines in SO₄²⁻ could elicit fast declines (i.e., ≤1−2 years) in MeHg concentrations[51]. Reducing agricultural S could be achieved as part of a sustainable strategy to manage S use[44], akin to nitrogen and phosphorus, which would also be expected to decrease wetland conditions of DOC concentrations, DOM aromaticity, and DOM thiol content[32−34], all of which decrease MeHg formation[19,40,41] and increase photo-demethylation of MeHg in surface waters[57]. Other local management strategies, including minimizing the drying and re-wetting of Everglades soils[69], may have beneficial outcomes by reducing the inadvertent oxidation of legacy S in peat to SO₄²⁻[45]. Changes in water delivery and water quality as part of local management and restoration activities may be effective in mitigating Hg methylation and other potential biogeochemical responses, such as DOM release, related to elevated SO₄²⁻ concentrations in the Everglades as well as other subtropical wetland systems.

Beyond local management options to mitigate Hg, decreases in atmospheric deposition of Hg stemming from national and international efforts are expected to potentially reduce the formation[43] and food web uptake of MeHg[70], but the timeline of a response on the landscape is uncertain. Across the conterminous US, the atmospheric deposition of Hg has declines over the last two decades due to decreased domestic emissions, but atmospheric deposition in south Florida has remained consistently elevated (1999−2023)[46] due to effective Hg oxidation in the troposphere in this region[2] and the air masses in southern Florida being distinct from the rest of the conterminous US. For these reasons, total Hg concentrations in water have been observed to be rather static in Everglades National Park (2008−2018)[48]. Declines in atmospheric deposition in South Florida are expected to occur in response to global reductions in Hg emissions, via the implementation of the United Nations' Minamata Convention on Mercury and reductions in carbon emissions that also decrease Hg releases[5]. Current scenario projections of global

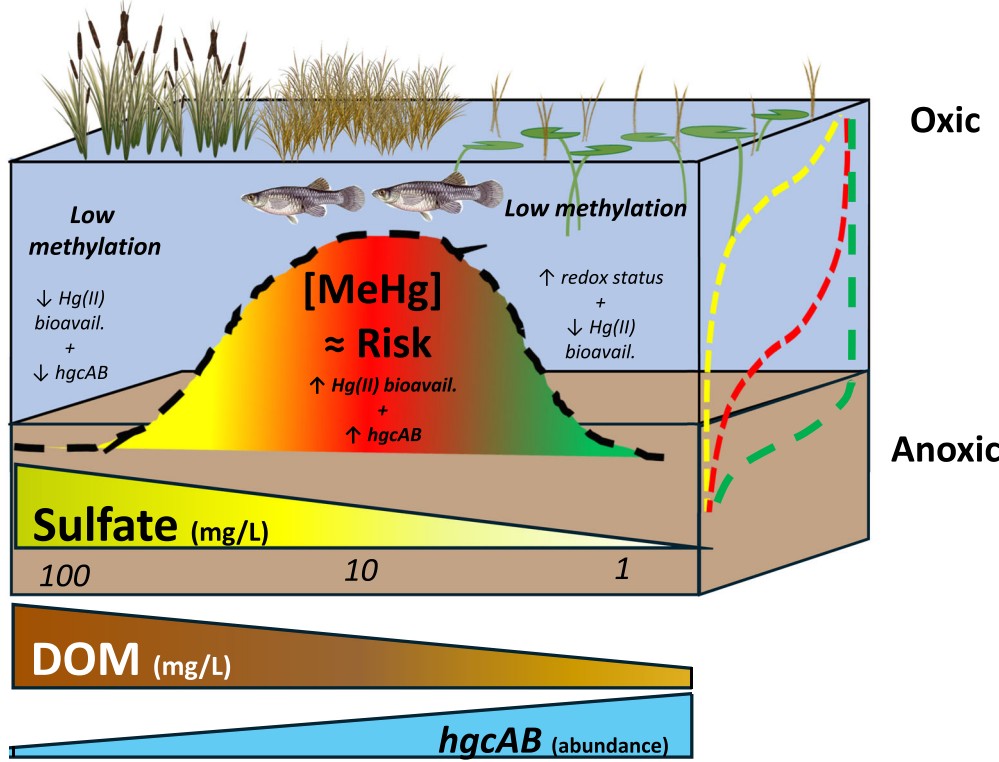

**Fig. 7 | Conceptual framework for sulfate (SO$_4^{2-}$) and dissolved organic matter (DOM) effects on methylmercury (MeHg) risk in subtropical wetlands.** Low risk is observed at high (->15 mg/L) and low (<1 mg/L) SO$_4^{2-}$ that facilitate unfavorable conditions for MeHg formation, due to geochemical bioavailability of Hg(II), wetland redox status, and *hgcAB* abundance of microbial community[19]. High risk is observed at intermediate SO$_4^{2-}$ (2–12 mg/L) that facilitates a suitable redox status, Hg(II) bioavailability (due to DOM aromaticity[36,40] and thiol content)[34,41], and *hgcAB* abundance[19] that collectively promote MeHg formation in proximity to the aquatic food web.

reductions[71] suggest that decreases in atmospheric Hg deposition will be modest over the coming three decades, with considerable uncertainty based on Hg reduction scenarios. However, despite the projected decreases in atmospheric Hg deposition, other geochemical and ecological factors may exert disproportionate control over ecosystem responses and lead to a range of potential outcomes that do not have a 1:1 response with declining Hg emissions and deposition[72,73]. Natural resource managers are tasked with balancing short- and long-term strategies to decrease MeHg formation and uptake in the aquatic food web, with SO$_4^{2-}$ reductions offering an attractive local approach that has other positive benefits on ecosystem health[44,45].

## Methods

### Field sampling locations

Field sites were selected to follow existing hydrologic and geochemical gradients across three freshwater, sawgrass-dominated wetlands of the Florida Everglades: water conservation areas 2A and 3A (WCA-2A, WCA-3A), and the Arthur R. Marshall Loxahatchee National Wildlife Refuge (LOX) (Fig. 1 and Supplementary Information (SI) Table S1). WCA-2A and WCA-3A receive high and intermediate concentrations of sulfate from canal inputs that drain upgradient agricultural lands, respectively, and generally exhibit decreasing sulfate concentrations from north to south[34,45]. Surface water flow vectors typically observed across WCA-2A and 3A are shown in Fig. 1, based on the Everglades Depth Estimation Network (EDEN) Surface-Water Interpolation Model Version 3[50]. The WCA-2A transect consisted of 16 sites over 14.5 km within the wetland that followed water flow south from the levee-6 (L-6) canal inputs to the middle of WCA-2A; one additional site of the L-6 canal immediately upgradient of the wetland was sampled to characterize the water

discharging into WCA-2A from the control structure. The WCA-3A transect consisted of 15 sites over 39.7 km, starting at the terminus of the levee-28 (L-28) canal and extending south to the intersection of WCA-3A and the L-29 canal. Due to lack of access to the L-28 canal within WCA-3A, the L-28 canal was sampled at the junction with highway I-75 to assess the composition of water entering WCA-3A (Fig. 1). LOX, in contrast to WCA-2A and WCA-3A, generally exhibits lower sulfate concentrations due to the interior regions being primarily rainfall-driven and hydrologically isolated from canal inputs by a perimeter canal[34,45]. However, contributions of canal-derived constituents (e.g., SO$_4^{2-}$, chloride (Cl$^-$)) intrude across the levees and penetrate the interior of LOX[45,74]. Five locations were sampled in LOX from the interior (LOX8) to the levee near the L-40 canal (LOX-136) (Fig. 1), which spanned 12.5 km. Sites across the three WCAs were previously evaluated to quantify the effects of SO$_4^{2-}$ on DOM reduced sulfur content and speciation[34] and microbial metagenomic sequencing for the Hg methylation genes (*hgcAB*+)[19].

### Water collection and analyses

Wetland and canal locations were sampled approximately once annually from 2012 to 2019, spanning from May to December. Table S2 summarizes the sampling date ranges, cumulative precipitation and wet Hg deposition data (on an annual basis and the 3 months prior to sampling) from local stations maintained by the National Atmospheric Deposition Program[46], and air temperature data during sampling. The cumulative annual precipitation and wet Hg deposition over the years of the study were within the range typically observed at these sites, based on available data records (1997–2023; Table S2). In total, 189 and 23 discrete water samples were collected from wetlands (WCA-2A, $n = 99$; WCA-3A, $n = 78$; LOX, $n = 12$)[49] and canals[75], respectively, across

the study period during seven field campaigns. The density of samples across the three wetlands aimed to (1) span the complete range of $SO_4^{2-}$ concentrations of the freshwater Florida Everglades[45] with comparable density as a function of $SO_4^{2-}$ concentration and (2) have higher sample density at the low-to-intermediate concentration range of $SO_4^{2-}$ (≤0.5–12 mg/L) (Fig. S4). At times, transects were sampled in a truncated design (Table S2), where some of the sites shown in Fig. 1 were skipped to best meet the abovementioned sampling goals.

At all sites, surface water was collected in 2 L polyethylene terephthalate (PETE) bottles at the air-water interface and stored in coolers on wet ice until processing the same day. For wetland sites only, pore water was collected 10 cm below the sediment-water interface at a rate of 100 mL min$^{-1}$ using a Teflon sipper connected to Teflon tubing and a peristaltic pump. The sipper was repositioned laterally by ~0.5 m every 10 min to not deplete sediment pore water. First, pore water temperature, conductivity (Orion four-cell conductivity electrode), pH (Orion ROSS Ultra™ electrode), dissolved oxygen (DO) concentration (Orion RDO optical probe), and oxidation-reduction potential (ORP; Orion ORP Triode electrode) were measured using a flow-through cell (Geotech; 40 mL dead volume) and multi-parameter meters (Orion Star™ A329, Beckman Coulter pHi 410)[56]. Next, pore waters were in-line filtered during collection (QFF, 0.7 µm pore size, pre-combusted at 550 °C, Whatman™) for the following analyses: sulfide (high-density polyethylene (HDPE) bottles, preserved with 50% volume/volume (v/v) sulfide antioxidant buffer), inorganic anions (HDPE bottles, no preservation, stored at 4°C), major cations anions (HDPE bottles, no preservation, stored at 4 °C), DOC concentration and DOM ultraviolet and visible (UV-vis) light absorption (pre-baked amber borosilicate glass vials at 450 °C for 4.5 h, no preservation, stored at 4 °C), and filtered total Hg (f.THg) and filtered MeHg (f.MeHg) (acid-cleaned Teflon bottles, preserved with 1% trace-metal grade hydrochloric acid). Pore water particulate total Hg (p.THg) and particulate MeHg (p.MeHg) concentrations were not measured due to potential artifacts during sample collection. Surface waters were filtered within 8 h of collection by vacuum filtration through a QFF for the identical set of analyses as pore water samples detailed above; QFF filters were frozen (−20 °C) for quantification of p.THg and p.MeHg concentrations. Field replicates of wetland surface waters and pore waters ($n = 8$) were sampled in 5% of total samples[49]. The average relative percent deviations of field replicates for concentrations of f.THg, f.MeHg, DOC, and $SO_4^{2-}$ were 5.5, 4.2, 0.6, and 2.8%, respectively.

Complete details on the analysis methods and limits of quantification are provided in an associated ScienceBase data release[49]. Sulfide was quantified within 12 h of sample collection by an ion-selective electrode. Major inorganic anions (Cl$^-$, nitrate (NO$_3^-$), $SO_4^{2-}$) were quantified by ion chromatography[76]. DOC concentration was quantified by persulfate oxidation (OI Analytical, model 700)[77]. UV-vis absorption spectra were measured from 190 to 800 nm, and decadic absorbance values were converted to absorption coefficients as

$$\alpha_\lambda = \frac{A_\lambda}{l} \tag{2}$$

where $\alpha_\lambda$ is the decadic absorption coefficient (cm$^{-1}$), $A_\lambda$ is the absorbance, and $l$ is the path length (cm). The DOM specific ultraviolet absorbance at 254 nm (SUVA$_{254}$), a proxy for DOM aromaticity[52] that indicates DOM source and reactivity to Hg(II)[19,36,40], was calculated as:

$$SUVA_{254} = \frac{\alpha_{254}}{[DOC]} \tag{3}$$

by dividing the $\alpha_{254}$ (m$^{-1}$) by DOC concentration (mg L$^{-1}$). Mercury measurements were made at the US Geological Survey Mercury

Research Laboratory (Madison, WI). THg measurements (f.THg, p.THg) were made by BrCl oxidation (0.2 M; 1% v/v for f.THg and 5% for p.THg), stannous chloride reduction, dual amalgamation, and cold vapor atomic fluorescence spectroscopy (CVAFS) detection following a modified version of US Environmental Protection Agency (EPA) method 1631, Revision E (Brooks-Rand TDM-II)[78]. Filtered and particulate MeHg were analyzed using a modified version of the US EPA method 1630[79] with isotope dilution. Approximately 20 pg of isotopically-enriched Me$^{199}$Hg and 1 mL of 1.6 M copper $SO_4^{2-}$ solution were added to 40 mL aliquots for MeHg analysis. Ambient MeHg was determined by relating the ratio of added Me$^{199}$Hg to Me$^{202}$Hg after distillation, aqueous phase ethylation, trapping on Tenax (Buchem B.V.), isothermal gas chromatography separation, and detection by inductively coupled plasma mass spectrometry (iCAP, Thermo Scientific) using an automated MeHg analyzer (MERX-M, Brooks-Rand). All field process blanks ($n = 15$) and quality assurance and quality control data are provided in the SI (Section S2; Fig. S12). The concentration of filtered Hg(II) (f.Hg(II)) was calculated as:

$$f.Hg(II) = f.THg - f.MeHg \tag{4}$$

In cases where the concentration of f.MeHg was below the daily detection limit ($n = 14$ instances of 189 wetland and canal samples)[49], the daily detection limit was used in the calculation of f.Hg(II). Dissolved gaseous Hg, reported at <0.04 ng L$^{-1}$ in the Everglades[3], was assumed to have minimal influence on calculated f.Hg(II) values.

### Fish collection and analyses
*Gambusia holbrooki* (termed gambusia) were collected from wetland sites using a 30 cm diameter Teflon sieve. Gambusia was selected because they have a short life span (≤6 months), feed on a mixture of periphyton and zooplankton, and reflect MeHg availability to the proximal aquatic food web in the recent past[17,43,48]. A composite sample was collected for each location, often consisting of 20–30 individual fish. Samples were frozen on-site (–80 °C), lyophilized, and homogenized via ball mill. Biological samples were first digested in 4.5 M Omnitrace nitric acid prior to biological MeHg analysis[80]. After the completion of MeHg analysis, samples were oxidized with bromine monochloride (BrCl, 10% v/v) and analyzed for total Hg by CVAFS using US EPA Method 1631, Revision E[78].

### Statistical analyses
Statistical comparisons between variables were assessed using linear regression analysis and paired *t*-tests (SigmaPlot v.14.5); *p* values <0.05 were considered statistically significant. For paired *t*-tests, a normality test (Shapiro–Wilk) was first performed. Data that passed the normality test were analyzed by a Welch's *t*-test, where equal variances are not assumed, and those that did not pass the normality tests were analyzed by the Mann–Whitney rank sum test (suitable for nonparametric data).

### Reporting summary
Further information on research design is available in the Nature Portfolio Reporting Summary linked to this article.

## Data availability
All water and fish data in this study have been deposited in the USGS ScienceBase data release product (https://doi.org/10.5066/P976EGIX).

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

## Acknowledgements

Financial support was provided to B.A.P., S.E.J., G.R.A. and D.P.K. by the US Geological Survey Greater Everglades Priority Ecosystems Science (GEPES) Program. We thank Eric Swain (USGS) for preparing vector flows from EDEN, Laura Flucke (USGS) for assistance with the GIS map, and Chris Eckley (US EPA) for helpful suggestions on the manuscript. We also thank the staff of the US Geological Survey Mercury Research Laboratory for field and analytical support. Any use of trade, firm, or product

names is for descriptive purposes only and does not imply endorsement by the US Government.

## Author contributions

B.A.P. contributed conceptualization, investigation, formal analysis, and writing—original draft. M.T.T. contributed conceptualization, methodology, investigation, formal analysis, data curation, and writing—review and editing. S.E.J. contributed formal analysis and writing—review and editing. G.R.A. contributed conceptualization and investigation. D.P.K. contributed conceptualization, investigation, project administration, and writing—review and editing.

## Competing interests

The authors declare no competing interests.
