## [Peer Review file · Nature Communications]

A Comprehensive Sulfate and DOM Framework to Assess Methylmercury Formation and Risk in Subtropical Wetlands

Corresponding Author: Dr Brett Poulin

Version 0:

Reviewer comments:

Reviewer #1

(Remarks to the Author)

This manuscript investigated role of sulfate loading on the methylation of Hg(II) in the surface water and pore water of the subtropical wetlands to predict its future response to the management of land and water as well as climate change affecting sulfate concentrations in wetland waters. For this purpose, sulfate, sulfide, SUVA, DOC, THg and MeHg concentrations were analyzed from the surface and pore water samples. The authors found that sulfate increases Hg(II) methylation in its low to intermediate levels through enhancing Hg(II) availability and Hg(II) methylating communities across the wetland length in the three represented zones. While experimental methods and data interpretation are solid and discussion is well described based on thorough literature review, my only concern is the novelty of this work, since the unimodal response of MeHg on sulfate concentration has been known for a while (lines 307-320). Revision suggestions are found below.

1. Lines 74-75: Suggest to add quantitative information on 'high and intermediate' concentration ranges.
2. Lines 91-92: "..... to those highly enriched in SO₄²⁻ (> 65 mg/L) in rheotrophic wetlands (?)." Suggest to add 'rheotrophic' or other words proper here.
3. Lines 145-146: Suggest to insert DO concentrations in Figure 2 or Supporting Information.
4. Lines 169-170: "therefore decreases in the molar SO₄/Cl are interpreted to be primarily due to microbial SO₄ reduction." Does it mean a decline of DO following the distance from the canal? Please clarify this vagueness.
5. Lines 182-185: "Higher DOC concentration and greater DOM aromaticity in wetland pore waters near canal inputs are interpreted to be the result of SO₄ stimulating degradation of peat by SO₄ reducing bacteria that releases DOM of higher aromaticity." Suggest to give example molecular groups that can be released to water via peat degradation.
6. Lines 210-211 "Across all wetlands, particulate Hg fractions were measured (p.Hg(II), p.MeHg) and were a minor fraction (accounted for ≤10% of the total Hg) (Figure S3)." However, Figure S3 does not show particulate Hg fraction.
7. Lines 251-257: A relatively large difference is observed between the measured and predicted MeHg concentrations in the low MeHg range (Figure S7a). The reason of this errors should be discussed.
8. Lines 251-257: Figure S7 (B) was not explained in the MS.
9. Lines 257-260: Why does this correlation (%MeHg/THg vs filtered MeHg) mean recent MeHg production? Suggest to give detailed explanation.
10. Lines 286-290: Aggregation of nano-HgS(s) was suggested as a reason for low Hg(II) methylation in the WCA-2A canal zone. How about the role of high DOM at the same site?
11. Lines 295-298: I agree with this interpretation, however, the role of redox potential cannot be ignored, since sulfate reduction rate and Hg(II) methylation rates are directly related to the anaerobic microbial activities.
12. Lines 321-326: Suggest to add a brief explanation on ecology of gambusia to justify the positive correlation between fish MeHg and surface water filtered MeHg concentration. What is the major carbon source of this species?
13. Lines 342: Wondering how climate change increases sulfate contraption in freshwater ecosystem.
14. Figure 7: HgcAB was not analyzed in this study. Suggest to include SO₄/Cl instead of HgcAB as it was used as a surrogate.
15. My only concern is the novelty of this work, since unimodal responses of MeHg on sulfate concentration have been known for a while, as described in the MS (lines 307-320). Please clearly emphasize the novelty and novelty of this study in the abstract or conclusion.

Reviewer #2

(Remarks to the Author)

General comments

This manuscript presents an original study that adds to the existing body of knowledge surrounding sulphate and methylmercury dynamics in wetlands and will certainly be of interest to the mercury research community. Though some polishing of language is necessary, the methods are clearly described and the manuscript is reasonable to follow. However, presumably to stay within the word limit, the authors exclude some details that would be helpful to ensure a reader's understanding. My comments are primarily requesting clarification, on these two fronts and other points detailed below: First, in the title and throughout the manuscript, the authors refer to the idea of 'methylmercury risk', but this is never clearly defined. Is risk equivalent to MeHg concentrations? In Fig 7, this seems to be implied, but also related to bioavailability and hgcAB abundance? High background [MeHg] could suggest that a system is less likely to experience increases due to perturbations, and therefore at lower 'risk'.

Second, the field methods are unclear without going into the SI. It appears that WCA-3A was sampled less than WCA-2A, and LOX less frequently than either. Differences in these sampling timelines are not explained, nor are the possible effects of sampling at different times throughout the years. This could be beneficial in terms of covering a wider range of flow conditions but also seems like it would be a confounding factor in the concentrations observed in fish, since they would be at a different stage in the life cycle.

Detailed comments

Abstract

Line 19: This is quite specific for the start of an abstract. Consider opening with a broader statement regarding MeHg in wetlands.

Lines 20-21: Consider 'land and water management strategies and climate change effects' for specificity.

Lines 22-23: Consider rearranging to 'We sampled surface and pore waters across SO₄ gradients of three freshwater wetlands in the Florida Everglades...'

Intro

Line 51: Here and throughout, rather than SO₄ 'levels', consider 'concentrations' or 'loads' for greater specificity.

Line 83: Here and throughout, land and water management is mentioned, but the introduction thus far only mentions agriculture. Please expand.

Results and Discussion

Line 111: Should say '...water conservation areas studied...'

Line 112: Comma should be after '(LOX))' in next line.

Lines 141-142: This is quite confusing; is there a way to rephrase without using 'distance from canals' twice?

Line 155: Here and throughout, there are comparisons such as 'notably lower', 'decreases', etc. These would benefit from support of some simple statistics.

Line 182: Similar or modestly higher aromaticity compared to surface water?

Line 191: The connection between photochemical transformation of DOM and autochthonous DOM production to increased distance from canals is unclear.

Line 222: Typo: 'qualitative'

Line 226: Consider expanding on the 'additional source and sink processes' or removing the statement.

Line 229: What does %MeHg tell us? E.g. what is it an indicator of that makes it distinct from [MeHg]?

Line 249: Typo?

Line 282: Typo: notably

Line 292: Typo: observe

Line 296: Not sure that the results of the study really support that SO₄ inputs 'indirectly alter the microbial metabolism'.

Lines 299-300: Consider adding climate summaries (e.g. average annual temperature, total precip) to SI to illustrate representativeness of data collected.

Line 316: 'SO₄ is a master variable' seems like an overconfident assertion, especially given the limited within-year temporal coverage of the study.

Lines 323-324: Consider also adding movement patterns throughout the life cycle to tie observations to locations. Given the short life span, did the varied sampling timeline influence the observations at all?

Line 361: Consider including estimated timeline of response for local interventions as well.

Methods

Consider adding a brief section detailing statistical analyses.

Line 380: Change 'following' to 'follow' and remove 'span'.

Line 383: How do WCA-2A and WCA-3A differ from each other? If they are very similar, what value does the second add?

Line 386: Figure caption says GARDEN rather than EDEN; presumably this is the same model under different names? If so, sticking with one would be better.

Line 388: in L-6, is L an acronym or abbreviation for something?

Line 404-405: Additional details are warranted here. For example, there looks to have been no sampling in 2016. WCA-3A was not sampled in 2012, and LOX was only sampled in 2015 and 2019. Is that correct? If so, the current description is quite misleading. Please also add reasoning for truncating transects in later years of the study.

Line 449: Please report Hg QA/QC info, at least in the SI.

Line 455: Consider adding reasoning for selecting this species for readers unfamiliar with the Everglades.

Version 1:

Reviewer comments:

Reviewer #1

(Remarks to the Author)

- 1) All the comments and suggestions were answered well.
- 2) Suggest to correct Figure 7 title for better reading: 'due to' was used four times.

Reviewer #2

(Remarks to the Author)

I am satisfied that my concerns have been addressed and have no further suggestions for revision.

Authors Note:

We have carefully considered the comments from the two reviewers and have revised the manuscript based on the reviewers' comments or provided detailed responses to the comments below. All responses are shown in blue text, and line numbers reference the Track Changes version of the revised manuscript. Revised text are shown indented and *italicized*. In addition to the two journal reviewer comments, the manuscript also underwent peer review by a federal employee (Dr. Chris Eckley, U.S. EPA), which is part of Fundamental Science Practices for manuscript approval by the U.S. Geological Survey. A few minor modifications were made in response to the review by Dr Eckley, which are shown in Track Changes in the revised manuscript. We appreciate the invitation to submit a revision of this manuscript for consideration. Thank you for your time.

Reviewer #1 (Remarks to the Author):

This manuscript investigated role of sulfate loading on the methylation of Hg(II) in the surface water and pore water of the subtropical wetlands to predict its future response to the management of land and water as well as climate change affecting sulfate concentrations in wetland waters. For this purpose, sulfate, sulfide, SUVA, DOC, THg and MeHg concentrations were analyzed from the surface and pore water samples. The authors found that sulfate increases Hg(II) methylation in its low to intermediate levels through enhancing Hg(II) availability and Hg(II) methylating communities across the wetland length in the three represented zones. While experimental methods and data interpretation are solid and discussion is well described based on thorough literature review, my only concern is the novelty of this work, since the unimodal response of MeHg on sulfate concentration has been known for a while (lines 307-320). Revision suggestions are found below.

We have made several modifications to the manuscript to highlight the novelty of the study, which focus on (1) the linkages established between geochemical and hydrological processes and MeHg formation and (2) this being the first study that links water quality conditions directly to MeHg formation and uptake in the aquatic food web. It is important to recognize that the observations reported are at the ecosystem scale (Figure 1), as they covered the three major freshwater wetlands of the Florida Everglades. These modifications to highlight the novelty were made in adherence to the *Nature Communications* Formatting Instructions, which state "Avoid phrases like "novel", "new", "for the first time", and "unprecedented" throughout the manuscript".

Abstract: Lines 32-35

"The study reports coherence between SO_4^{2-} and MeHg concentrations and biologic uptake that permits an advancement in MeHg risk assessment to aquatic food webs that incorporate geochemical, microbial, and hydrological wetland processes, which has

global implications due to the threat of increased SO_4^{2-} from anthropogenic sulfur uses and climate change.”

Introduction: Lines 57-58 & 108-113

“A revisitation of linkages between SO_4^{2-} concentrations and the potential for MeHg formation and uptake in biota is needed.”

“The findings advances a new ecosystem-scale conceptual model of the effects of SO_4^{2-} on Hg risk to the aquatic food web and are discussed in context of parallel efforts that quantified the effects of SO_4^{2-} on DOM thiol content,³⁴ the abundance and metabolic capabilities of hgcAB+ microorganisms,¹⁹ downgradient impacts on MeHg formation and biouptake in Everglades National Park,⁴⁸ and strategies to mitigate the risk of Hg to wildlife and humans.”

Results & Discussion: Lines 343-346, 351-357, 391-395

“The field data underlying these biogeochemical relationships span multiple years (2012-2019) and seasons (May and December) (Table S2) and leverage geochemical,³⁴ hydrological, and microbial insights,¹⁹ providing a comprehensive dataset to develop a framework for MeHg production and risk.”

“The unimodal relationship here between SO_4^{2-} and f.MeHg concentration (Figures 3b, 5) is consistent with the conceptual model first proposed by Gilmour and Henry (1994)⁸ but incorporates recent innovations in the understanding of Hg(II) nano-scale geochemistry and the complexities of microbial Hg(II) methylation, which provides mechanistic insights that explain MeHg formation across the greater Everglades¹⁶ and in response to increases⁴³ and decreases in SO_4^{2-} loading.⁵¹”

“The comprehensive framework presented here (Figure 7) for the Florida Everglades support that efforts to decrease SO_4^{2-} concentrations from agricultural sources have promise to decrease aqueous and biological MeHg concentrations in wetlands (Figures 5, 6), offering a local strategy to mitigate Hg contamination.”

1. Lines 74-75: Suggest to add quantitative information on ‘high and intermediate’ concentration ranges.

We have added clarification on the sulfate concentration ranges defined as low, intermediate, and high, at a later portion of the manuscript (Lines 96-98, 462-465; see new Figure S4), and simplified the presentation here in the Introduction.

Lines 85-88

“The combination of SO_4^{2-} effects on Hg(II) bioavailability for methylation and activity of the microbial community are understood to contribute to the non-linear response observed between SO_4^{2-} and MeHg,^{8,9,11-15,43}”

2. Lines 91-92: “..... to those highly enriched in SO_4^{2-} (> 65 mg/L) in rheotrophic wetlands (?)” Suggest to add ‘rheotrophic’ or other words proper here.

We appreciate the suggestion from the reviewer. However, the terminology of “rhoetrophic” is a regional-specific term of a wetland with continuous high-water table. We have chosen to forego adding a specific term here, but have modified the presentation to clarify that the wetlands have standing water year round (Lines 96-98).

“The wetland transects exhibited distinct horizontal geochemical gradients in SO_4^{2-} spanning from low SO_4^{2-} ($< 0.5 \text{ mg/L}$) wetlands that receive SO_4^{2-} solely from the atmosphere to those highly enriched in SO_4^{2-} ($> 65 \text{ mg/L}$) from agricultural runoff.^{44,46}”

The following statement in the methods section was also clarified to ensure the reader understands the flow vectors are for surface flow (Lines 434-436)

*“Surface water flow vectors typically observed across WCA-2A and 3A are shown in **Figure 1**, based on the Everglades Depth Estimation Network (EDEN) Surface-Water Interpolation Model Version 3.⁴⁹”*

3. Lines 145-146: Suggest to insert DO concentrations in Figure 2 or Supporting Information.

We appreciate the suggestion and have added DO and ORP data to the SI in a new figure (Figure S3). We have also revised the presentation to further the comparison to include both DO and ORP, two important metrics that differ in response to sulfate.

Figure S3. Measurements of (a-c) oxidation-reduction potential (ORP) and (d-f) dissolved oxygen (O₂) of surface and pore waters from Water Conservation Area 2A (WCA-2A) (August, 2014), WCA-3A (August, 2014), and the Arthur R. Marshall Loxahatchee National Wildlife Refuge (LOX) as a function of distance from canal inputs (November, 2015). Dashed lines are presented to guide the eye.

Lines 153-160

*“Further, surface waters were sub-oxic to anoxic in all three wetlands immediate downgradient of canal inputs (**Figure S3a-c**), with concentrations generally increasing with distance from canal inputs.⁴⁹ The oxidation-reduction potential (ORP) of wetlands*

surface water and pore waters (Figure S3d-f) reflected the degree of canal influence on wetland redox status, with significantly lower ORP in pore waters of WCA-2A (-271 ± 12.3 mV; average \pm std) compared to WCA-3A (-168.7 ± 18.4 mV) or the interior of LOX (LOX-8, -25 mV) (Welch's t-test; $p < 0.001$)."

4. Lines 169-170: "therefore decreases in the molar SO_4/Cl are interpreted to be primarily due to microbial SO_4 reduction." Does it mean a decline of DO following the distance from the canal? Please clarify this vagueness.

Please see our response to the comment above that expanded the study to include DO concentration data.

5. Lines 182-185: "Higher DOC concentration and greater DOM aromaticity in wetland pore waters near canal inputs are interpreted to be the result of SO_4 stimulating degradation of peat by SO_4 reducing bacteria that releases DOM of higher aromaticity." Suggest to give example molecular groups that can be released to water via peat degradation.

We have added an example of a type of DOM molecular group that would be released, which is from the mechanistic studies on this topic. (Lines 201-204)

"Higher DOC concentration and greater DOM aromaticity in wetland pore waters near canal inputs compared to surface waters are interpreted to be the result of SO_4^{2-} stimulating degradation of peat by SO_4^{2-} reducing bacteria³³ that releases DOM of higher aromaticity (e.g., phenolic groups).^{29,32}"

6. Lines 210-211 "Across all wetlands, particulate Hg fractions were measured (p.Hg(II), p.MeHg) and were a minor fraction (accounted for $\leq 10\%$ of the total Hg) (Figure S3)." However, Figure S3 does not show particulate Hg fraction.

We appreciate the reviewer identifying the incorrect Figure call out. We have revised the text to properly call out Figure S1. We also revised Figure S1 to show the particulate percentage (rather than the filtered percentage) to improve clarity.

Figure S1. Box plots present median and quartile ranges of percentage of (a) total total Hg as particulate HgT and (b) total MeHg as particulate MeHg between surface waters of Water Conservations Areas (WCA-2A, 3A, LOX) and canals that drain into these wetlands (L-6 and L-28 canals). Error bars represent 10-90% percentiles, outliers are shown as data points, and p-values present results from a paired t-tests between WCA and canal data.

7. Lines 251-257: A relatively large difference is observed between the measured and predicted MeHg concentrations in the low MeHg range (Figure S7a). The reason of this errors should be discussed.

We have expanded the discussion to detail the discrepancy and include an explanation of why there is higher differences between measured and predicted MeHg concentrations at low environmental MeHg

Lines 286-289

*“At both high (>30 mg/L) and low SO_4^{2-} concentration (<1 mg/L), which corresponded to low f.MeHg concentration (< 0.1 ng/L; **Figure S8a**), a modest discrepancy was noted between observed and predicted f.MeHg concentrations.”*

Lines 310-313

*“The modest disagreement between observed and predicted f.MeHg at SO_4^{2-} concentrations < 1 mg/L (**Figure 3d**) simply reflects the low MeHg formation under low, ambient levels of SO_4^{2-} that are not well fit using a 3-parameter unimodal function.”*

8. Lines 251-257: Figure S7 (B) was not explained in the MS.

We added a sentence to the main text explaining the use of residual f.MeHg as an assessment of the unimodal equation fit. (Lines 282-286)

“There was good agreement between observed and predicted f.MeHg concentrations at observed f.MeHg concentrations ≥ 0.1 ng/L (Figure S7a) with no systematic difference in residual f.MeHg concentrations (i.e., the difference between observed and predicted f.MeHg concentrations) across the range of SO_4^{2-} concentration (Figure S7b), indicating the unimodal fit was robust across the conditions of the three wetlands.”

9. Lines 257-260: Why does this correlation (%MeHg/THg vs filtered MeHg) mean recent MeHg production? Suggest to give detailed explanation.

We have expanded this paragraph to better explain why high % MeHg and high concentrations of MeHg are supportive of recent production. The reasons are that higher MeHg concentration could be due to more inorganic Hg(II) that is available to methylated, but an increase in both MeHg concentration and the percentage of total Hg as MeHg can only be explained by greater local production. (Lines 291-296)

“Further, across all wetlands, the %MeHg was significantly correlated ($R^2=0.77$, $p<0.001$; $n=168$) to the concentration of f.MeHg in water (Figure S8), which has been observed in the Everglades^{9,19} and other systems^{10,15} to correspond with faster measured rates of MeHg production. Thus, wetlands with higher aqueous f.MeHg and %MeHg are interpreted to reflect recent MeHg production, as MeHg that is produced is susceptible to sink processes (e.g., photo-demethylation,⁵⁷ adsorption to peat).⁵⁸”

10. Lines 286-290: Aggregation of nano-HgS(s) was suggested as a reason for low Hg(II) methylation in the WCA-2A canal zone. How about the role of high DOM at the same site?

We expanded the discussion to discuss this point in greater depth. (Lines 331-334)

“Despite the elevated concentrations of highly aromatic, sulfurized DOM³⁴ under these sulfidic conditions, crystalline nano- β -HgS(s)³⁶ will form and dramatically decrease f.Hg(II) concentrations in the wetland pore and surface waters (Figures 3c, 4a) and contributes to decreased MeHg formation.⁶⁴”

11. Lines 295-298: I agree with this interpretation, however, the role of redox potential cannot be ignored, since sulfate reduction rate and Hg(II) methylation rates are directly related to the anaerobic microbial activities.

We have included data in Figure S3 with oxidation reduction potential (ORP) field measurements, and amended the text that used statistics to show significant differences between the three wetlands. Directly linking ORP to microbial activity is not feasible, however, as the latter is governed by factors controlling microbial activity (e.g., terminal electron acceptor availability, reduced carbon availability), whereas the former is a measurement of potential of the system based on various redox couples. Thus, we appreciate the suggestion of the reviewer

to include the ORP data, and these data strengthen the presentation of how sulfate directly influences redox state of the wetlands, but we cannot expand the presentation to link the redox potential to rates of microbial activity.

Lines 157-160

“The oxidation-reduction potential (ORP) of wetlands surface water and pore waters (Figure S3d-f) reflected the degree of canal influence on wetland redox status, with significantly lower ORP in pore waters of WCA-2A (-271 ± 12.3 mV; average \pm std) compared to WCA-3A (-168.7 ± 18.4 mV) or the interior of LOX (LOX-8, -25 mV) (Welch’s t-test; $p < 0.001$).”

Lines 223-227

“In summary, canal water SO_4^{2-} inputs lower the wetland redox status and stimulate S(–II) production and the release of aromatic DOM across distinct lateral and vertical geochemical gradients within all three Everglades wetlands, and subsequently these biogeochemical changes govern the geochemical nature of Hg(II)^{31,36} and bioavailability for methylation.^{39,40}”

Details on the collection of ORP and DO data have been added to the Methods section. We chose to include this in the main text, as Reviewer #2 highlighted the need for all methods details in the main text for clarity.

Lines 474-476

“First, pore water temperature, conductivity (Orion four-cell conductivity electrode), pH (Orion ROSS Ultra™ electrode), dissolved oxygen (DO) concentration (Orion RDO optical probe), and oxidation-reduction potential (ORP; Orion ORP Triode electrode) were measured using a flow-through cell (Geotech; 40 mL dead volume) and multi-parameter meters (Orion Star™ A329, Beckman Coulter pHi 410).⁵⁶”

12. Lines 321-326: Suggest to add a brief explanation on ecology of gambusia to justify the positive correlation between fish MeHg and surface water filtered MeHg concentration. What is the major carbon source of this species?

We have added information in several locations to justify the use of gambusia for this study, their life span, and primary carbon sources.

Lines 105-108

*“Lastly, relationships between MeHg concentrations in surface waters were compared to MeHg concentrations in prey fish (*Gambusia holbrooki*), an indicator species with a short life span (≤ 6 months) that reflects recent MeHg risk to the aquatic food web.^{17,47}”*

Lines 523-525

*“*Gambusia* was selected because they have a short life span (≤ 6 months), feed on a mixture of periphyton and zooplankton, and therefore will reflect MeHg availability to the proximal aquatic food web in the recent past.^{17,47}”*

The original content in the manuscript contained information on the carbon source in the paragraph noted by the reviewer (Line 346).

13. Lines 342: Wondering how climate change increases sulfate contraction in freshwater ecosystem.

We presume the reviewer is suggesting that these climate processes result in “contraction” of freshwater ecosystems. We modified the sentence in question to use this phrase.

Lines 389-391

“For example, enhanced mineral weathering in the Arctic has increased SO_4^{2-} concentrations in the Mackenzie River by 45% in the last 60 years²⁸ and sea-level rise is threatening coastal wetlands with marine SO_4^{2-} that will result in the contraction of freshwater ecosystems.²⁹”

14. Figure 7: HgcAB was not analyzed in this study. Suggest to include SO_4/Cl instead of HgcAB as it was used as a surrogate.

While the reviewer is correct, we did measure hgcAB at the same time as samples collected in the present study and those results have been published. Microbial hgcAB measurements were collected at sites that spanned the entire sulfate gradient documented in this study (across all three wetlands), and at the same depth as pore water samples collected here. Therefore, we have strong rationale to assist in interpretation of the present study. In the figure caption, we have cited the source of this interpretation (to clarify this to the reader), and at numerous locations throughout the manuscript we state that the present study leverages microbial insights gained from Peterson et al. (2023) (Lines 111, 308, 331, 449-451). No other changes were made.

15. My only concern is the novelty of this work, since unimodal responses of MeHg on sulfate concentration have been known for a while, as described in the MS (lines 307-320). Please clearly emphasize the novelty and novelty of this study in the abstract or conclusion. Please see our modification listed above on the novelty of the study (below the general comment by Reviewer #1).

Reviewer #2 (Remarks to the Author):

General comments

This manuscript presents an original study that adds to the existing body of knowledge surrounding sulphate and methylmercury dynamics in wetlands and will certainly be of interest to the mercury research community. Though some polishing of language is necessary, the methods are clearly described and the manuscript is reasonable to follow. However, presumably to stay within the word limit, the authors exclude some details that would be helpful to ensure a reader's understanding. My comments are primarily requesting clarification, on these two fronts and other points detailed below:

First, in the title and throughout the manuscript, the authors refer to the idea of ‘methylmercury risk’, but this is never clearly defined. Is risk equivalent to MeHg concentrations? In Fig 7, this

seems to be implied, but also related to bioavailability and hgcAB abundance? High background [MeHg] could suggest that a system is less likely to experience increases due to perturbations, and therefore at lower 'risk'.

We appreciate the suggestion for clarity by the reviewer. We have added statements in the Introduction and Results & Discussion to define MeHg "risk".

Lines 48-50

"MeHg risk is defined as the potential for MeHg to be formed within or proximal to aquatic systems that results in efficient MeHg assimilation in the aquatic food web."

Lines 373-375

"Figure 7 synthesizes a framework on the association between SO_4^{2-} , DOM, and the potential for MeHg formation in proximity to aquatic food webs, which dictates MeHg risk."

Second, the field methods are unclear without going into the SI. It appears that WCA-3A was sampled less than WCA-2A, and LOX less frequently than either. Differences in these sampling timelines are not explained, nor are the possible effects of sampling at different times throughout the years. This could be beneficial in terms of covering a wider range of flow conditions but also seems like it would be a confounding factor in the concentrations observed in fish, since they would be at a different stage in the life cycle.

It is unclear to the authors what the reviewer is referring to, in stating the "field methods are unclear without going to the SI". There are no methods presented in the SI (aside from detailed QA/QC methods that were added based on comments from Reviewer #2), as all the methods are presented in the main text. We have revised the methods presentation to expand on the justification for sampling the various wetlands at different intervals.

The reviewer asked for more extensive discussion on the role of seasonality, which was already included in the original submission (see underlined text shown below). The text shown below discusses the potential role of season, but given the many variables (time of year, calendar year, wetland location, season), the data are best explained by the explanatory variable of sulfate concentration (Figure 3d, Figure 5) which highlights that those other factors are secondary. No other changes were made.

Lines 343-351

"The field data underlying these biogeochemical relationships span multiple years (2012-2019) and seasons (May and December) (Table S2), providing a comprehensive dataset to develop a framework for MeHg production and risk. Taken together, a highly consistent trend is observed across all wetland surface waters between concentrations of SO_4^{2-} , the DOM α_{254} (which scales positively with increases in DOC concentrations and DOM SUVA₂₅₄), and f.MeHg concentration (Figure 5). We interpret the spread in f.MeHg at a given SO_4^{2-} concentration (Figures 3d, 5) to be the effects of seasonality on MeHg formation, as noted previously,^{4,17} with higher observed MeHg formation in the spring and summer months (Figure S10) due to higher rates of microbial metabolism and regular rainfall delivering Hg(II)³ to wetlands."

Detailed comments

Abstract

Line 19: This is quite specific for the start of an abstract. Consider opening with a broader statement regarding MeHg in wetlands.

We have added a short introductory sentence to the Abstract (Lines 16-17)

“Wetlands are critical environments for the biogeochemical cycling and ecological uptake of contaminants.”

Lines 20-21: Consider ‘land and water management strategies and climate change effects’ for specificity.

Accepted.

Lines 22-23: Consider rearranging to ‘We sampled surface and pore waters across SO₄ gradients of three freshwater wetlands in the Florida Everglades...’

Intro

Accepted

Line 51: Here and throughout, rather than SO₄ ‘levels’, consider ‘concentrations’ or ‘loads’ for greater specificity.

Here and throughout the entire manuscript, “levels” has been revised to “concentrations”, when referencing sulfate and any other constituent. We thank the reviewer for this suggestion to improve clarity in presentation.

Line 83: Here and throughout, land and water management is mentioned, but the introduction thus far only mentions agriculture. Please expand.

We have expanded this sentence to include more examples of water management actions. (Lines 88- 93)

“Given the potential for fluctuations in SO₄²⁻ concentrations in wetlands in response to land management (e.g., agricultural practices), water management (e.g., storm water treatment, phosphate mitigation via alum additions),^{43,44} or climate change processes,²⁷⁻²⁹ as well as the uncertainties in the geochemical and microbial responses pertaining to MeHg formation, ecosystem-scale studies are needed to solidify linkages between SO₄²⁻, MeHg formation, and subsequent MeHg uptake in aquatic food webs.”

Results and Discussion

Line 111: Should say ‘...water conservation areas studied...’

Accepted

Line 112: Comma should be after '(LOX))' in next line.

Accepted

Lines 141-142: This is quite confusing; is there a way to rephrase without using 'distance from canals' twice?

Revised. Lines 150-153.

“With increased distance from canals, surface water SO_4^{2-} concentrations consistently decreased (e.g., in August 2014, SO_4^{2-} decreased from 65.8 to 50.6 mg/L, 22.5 to <0.5 mg/L, and 4.1 to <0.5 mg/L across transects in WCA-2A, LOX, and WCA-3A, respectively).”

Line 155: Here and throughout, there are comparisons such as 'notably lower', 'decreases', etc. These would benefit from support of some simple statistics.

We have added a section to the Methods with the statistical approaches used, and expanded the statistical tests across populations of data or between variables. There are instances where “qualitative” language is retained the manuscript, as standard statistical tests were not suitable for portions of the data.

Modifications include:

Lines 171-173

*“First, in sediment pore waters, SO_4^{2-} concentrations were significantly lower than corresponding surface waters (Mann-Whitney Rank Sum Test, $p < 0.001$), by ≤ 60.5 , ≤ 26.5 , and ≤ 3.5 mg/L in WCA-2A, LOX, and WCA-3A, respectively (**Figures 2a-2c**).”*

Lines 198-201

*“In wetland pore waters, however, DOC concentrations were significantly higher compared to surface waters (Mann-Whitney Rank Sum Test, $p < 0.001$, $n = 70$), particularly in regions with active SO_4^{2-} reduction near canal inputs (**Figure 2j-2l**), and DOM in pore waters was of similar or modestly higher aromaticity (DOM SUVA_{254} ; Welch's t-test, $p < 0.001$, $n = 70$; **Figure 2m-2o**).”*

Lines 260-263

“Further downgradient, average f.MeHg concentrations and %MeHg were higher in surface waters (0.27 ± 0.05 ng/L and 16.4 ± 2.3 %MeHg, respectively; ± 1 standard deviation) compared to pore waters (0.06 ± 0.04 ng/L and 4.5 ± 2.8 %MeHg, respectively) or the L-6 canal (Mann-Whitney Rank Sum Test, $p < 0.001$, $n = 14$).”

We have added a section to the methods that details the statistical approaches used. Lines 530-535.

“Statistical comparisons between variables were assessed using linear regression analysis and paired t-tests (SigmaPlot v.14.5); p-values < 0.05 were considered of statistical significance. For paired t-tests, a normality test (Shapiro-Wilk) was first performed. Data that passed the normality test were analyzed by a Welch’s t-test, where equal variances are not assumed, and those that did not pass the normality tests were analyzed by Mann-Whitney Rank Sum Test (suitable for non-parametric data).”

Line 182: Similar or modestly higher aromaticity compared to surface water?

Accepted.

Lines 201-204

“Higher DOC concentration and greater DOM aromaticity in wetland pore waters near canal inputs compared to surface waters are interpreted to be the result of SO_4^{2-} stimulating degradation of peat by SO_4^{2-} reducing bacteria³³ that releases DOM of higher aromaticity (e.g., phenolic groups).^{29,32”}

Line 191: The connection between photochemical transformation of DOM and autochthonous DOM production to increased distance from canals is unclear.

We have revised the statement for clarity. Lines 207-211.

“With increased distance from canals in WCA-2A and WCA-3A, surface water DOC concentrations decreased (e.g., from 37.0 to 32.7 and 23.1 to 15.0 mg/L, respectively) and DOM SUVA₂₅₄ decreased (e.g., from 3.7 to 3.1 and 3.9 to 2.4 L/mg m, respectively) (Figure 2m, 2n), likely the result of the photochemical transformation of DOM⁵⁴ and autochthonous DOM production in wetland surface waters.^{55”}

Line 222: Typo: ‘qualitative’

“qualitative” was removed from the sentence in question.

Line 226: Consider expanding on the ‘additional source and sink processes’ or removing the statement.

We have expanded this statement to include the one other sink process, which is adsorption of Hg(II) to peat. The revised list of “source and sink processes” is complete, as it covers the major processes that will influence the instantaneous concentration of Hg(II) at any location.

Lines 247-250

“Although additional source and sink processes also influence f.Hg(II) concentrations in Everglades wetlands, including rainfall delivery of f.Hg(II),^{2,45} photo-reduction of Hg(II) to Hg(0),^{3,4} and partitioning of Hg(II) to peat,⁴⁷ SO_4^{2-} exhibits notable control on the concentration of f.Hg(II), which can limit MeHg formation in wetlands.”

Line 229: What does %MeHg tell us? E.g. what is it an indicator of that makes it distinct from [MeHg]?

We appreciate the suggestion to improve clarity in presentation. We have revised this to provide rationale up front in this section on the use of %MeHg (shown below, Lines 253-255) and in Lines 291-293 we detail how we interpret these results in context of previous studies.

Lines 253-255

*“The %MeHg term is used to normalize f.MeHg concentration data to the total Hg concentration at a given location, as f.Hg(II) concentrations varied by an order of magnitude within and between the studied wetlands (e.g., **Figures 4a-c, S6a-b**) and is the precursor for f.MeHg.^{9,55}”*

Lines 291-293

*“Across all wetlands, the %MeHg was significantly correlated ($R^2=0.77$, $p<0.001$; $n=168$) to the concentration of f.MeHg in water (**Figure S10**), which has been observed in the Everglades^{9,19} and other systems^{10,15} to correspond with faster measured rates of MeHg production.”*

Line 249: Typo?

It is unclear to us what is potentially a typo in this statement. We reviewed it and deemed it to be grammatically correct.

Line 282: Typo: notably

Accepted. We thank the reviewer for identifying this typo.

Line 292: Typo: observe

Accepted. We thank the reviewer for identifying this typo.

Line 296: Not sure that the results of the study really support that SO₄ inputs ‘indirectly alter the microbial metabolism’.

We are confident in this interpretation, as it is informed by genome-resolved metagenomics, which was able to identify the metabolic capabilities of the microorganisms with the hgcAB gene pair. No changes were made.

Lines 299-300: Consider adding climate summaries (e.g. average annual temperature, total precip) to SI to illustrate representativeness of data collected.

We have added information to Table S2 that includes the temperature at the time of sampling, accumulative precipitation and total Hg deposition (both annually and over the three month

period prior to sampling), and annual data from 1997-2023. The Methods section was revised to make this clear to the reader.

Lines 455-460.

“Table S2 summarizes the sampling date ranges, cumulative precipitation and wet Hg deposition data (on an annual basis and the 3 months prior to sampling) from local stations maintained by the National Atmospheric Deposition Program,⁴⁶ and air temperature data during sampling. The cumulative annual precipitation and wet Hg deposition over the years of the study were within the range typically observed at these sites, based on available data record (1997-2023; Table S2).”

Line 316: ‘SO4 is a master variable’ seems like an overconfident assertion, especially given the limited within-year temporal coverage of the study.

The reviewer is mistaken, claiming that the study is a “limited, within-year temporal coverage”. As we detail, the data were collected over seven field campaigns spanning 8 calendar years, and between the calendar months of May to December (e.g., as detailed in Table S2). Therefore, we do not agree that the statement is an overconfident assertion, as all the field data show coherence between the role of sulfate and MeHg concentration (Figure 3d, Figure 5). No changes were made.

Lines 323-324: Consider also adding movement patterns throughout the life cycle to tie observations to locations. Given the short life span, did the varied sampling timeline influence the observations at all?

As shown in Figure 6, across all wetlands (regardless of sampling time), there was a strong correlation between surface water MeHg and fish. Although it would be an interesting consideration to expand the discussion of the paper to include the movement patterns of gambusia, it is beyond the scope of this study and not feasible with the available data.

Revised text in the Methods outlines why we selected these fish and cites other studies that used them as indicator fish. No other changes were made.

Lines 523-525

“Gambusia were selected because they have a short life span (≤ 6 months), feed on a mixture of periphyton and zooplankton, and reflect MeHg availability to the proximal aquatic food web in the recent past.^{17,43,48”}

Line 361: Consider including estimated timeline of response for local interventions as well.

The original manuscript highlighted an estimated response to local interventions ($\leq 1-2$ years), but we have revised the statement for clarity.

Lines 395-397

“Local reductions in SO₄²⁻ concentration or load may yield a relatively fast response, as previous work in the Everglades demonstrated that declines in SO₄²⁻ could elicit rapid reductions (i.e., ≤ 1-2 years) in MeHg concentrations.⁵¹”

Methods

Consider adding a brief section detailing statistical analyses.

Accepted. See response above.

Line 380: Change ‘following’ to ‘follow’ and remove ‘span’.

Accepted. We thank the reviewer for identifying this typo.

Line 383: How do WCA-2A and WCA-3A differ from each other? If they are very similar, what value does the second add?

We have clarified that WCA-2A and 3A how high and intermediate concentrations of sulfate, or agricultural influence, which is why they were selected.

Lines 432-434

“WCA-2A and WCA-3A receive high and intermediate concentrations of sulfate from canal inputs that drain upgradient agricultural lands, respectively, and generally exhibit decreasing sulfate concentrations from north to south.^{34,44}”

Line 386: Figure caption says GARDEN rather than EDEN; presumably this is the same model under different names? If so, sticking with one would be better.

We appreciate the suggestion and have revised the main text and captions to only use “Everglades Depth Estimation Network (EDEN)” (Lines 162, 436, 760)

Line 388: in L-6, is L an acronym or abbreviation for something?

We have revised the presentation to clarify that the “L” stands for levee, as that is the control structure. This is defined at first presentation (Line 122, 125) and in the methods section (Line 438, 441) for all canals.

Line 404-405: Additional details are warranted here. For example, there looks to have been no sampling in 2016. WCA-3A was not sampled in 2012, and LOX was only sampled in 2015 and 2019. Is that correct? If so, the current description is quite misleading. Please also add reasoning for truncating transects in later years of the study.

Correct, we did not sample wetlands in 2016, which was due to unexpected health circumstances of one of the co-PIs of the project (George Aiken).

We have added additional details to justify the sampling plan across the wetlands in the methods and when introducing the sampling plan. The reason to sample “truncated” transects was to (1) sample the complete range of sulfate concentrations documented in freshwater wetlands of the Florida Everglades with approximately uniform sample density over the intermediate to high sulfate concentration range (12 – 72 mg/L), (2) and have higher sample density at the low-to-intermediate concentration range of sulfate (0-12 mg/L), which has been documented to result in the largest differences in MeHg in a variety of ecosystems. The histogram below shows the distribution of surface water sulfate concentrations in this study, which has been added to the SI (new Figure S4). The reviewer is correct that wetlands were samples in different density, but the goals of the study were to use the three wetlands to cover the entire gradient, which was achieved.

Figure S4. A histogram sulfate (SO₄²⁻) concentrations in the three wetlands of the study, with an sampling emphasis on low-to-intermediate concentrations (0-12 mg/L) and equal distribution of samples at intermediate-to-high SO₄²⁻ concentration (12-72 mg/L).

Lines 162-167

*“The Florida Everglades is naturally a low SO₄²⁻ environment, with concentrations between 0.1-1.0 mg/L in regions unimpacted by agricultural S inputs,⁴⁴ and therefore SO₄²⁻ concentrations observed in interior WCA wetlands greatly exceeded background concentrations (up to ~70-fold) and were distributed across the entire range observed (≤0.5 – 72.0 mg/L; **Figure S4**), creating vastly different water quality conditions across downgradient freshwater wetlands.”*

Lines 462-467

“The density of samples across the three wetlands aimed to (1) span the complete range of sulfate concentrations of the freshwater Florida Everglades⁴⁴ with comparable density as a function of SO_4^{2-} concentration and (2) have higher sample density at the low-to-intermediate concentration range of sulfate ($\leq 0.5\text{-}12$ mg/L) (**Figure S4**). At times, transects were sampled in a truncated design (**Table S2**), where some of the sites shown in Figure 1 were skipped to best meet the abovementioned sampling goals.”

Line 449: Please report Hg QA/QC info, at least in the SI.

We have included all QA/QC information for the analysis of total and methylmercury in the SI (Section S2), and included a figure with summary statistics for field process blanks. This is referenced in the main text methods.

Main Text Lines 513-514

“All field process blanks and quality assurance and quality control data are provided in the SI (**Figure S12**).”

SI Content: See SI Section S2: Quality Assurance and Quality Control on Mercury Measurements (SI Lines 48-76

Figure S12. A comparison of concentrations of field process blanks (n=15) and environmental samples from the study wetlands including (a) particulate total Hg (p.THg) (n=126), (b) particulate methylmercury (p.MeHg) (n=126), (c) filtered total Hg (f.THg) (n=197), and (d) filtered methylmercury (f.MeHg) (n=197). The horizontal dashed gray line presents the method detection limit.

Line 455: Consider adding reasoning for selecting this species for readers unfamiliar with the Everglades.

Please see our response to Reviewer#1, comment #12.

R1: Suggest to correct Figure 7 title for better reading: 'due to' was used four times.

We have revised Figure 7 caption to improve clarity and reduce redundant phrases.

Figure 7. Conceptual framework for sulfate (SO_4^{2-}) and dissolved organic matter (DOM) effects on methylmercury (MeHg) risk in subtropical wetlands. Low risk is observed at high ($\sim >15$ mg/L) and low (<1 mg/L) SO_4^{2-} that facilitate unfavorable conditions for MeHg formation, due to geochemical bioavailability of Hg(II), wetland redox status, and *hgcAB* abundance of microbial community.¹⁹ High risk is observed at intermediate SO_4^{2-} (3-12 mg/L) that facilitate suitable redox status, Hg(II) bioavailability (due to DOM aromaticity^{36,40} and thiol content),^{34,41} and *hgcAB* abundance¹⁹ that collectively promote MeHg formation in proximity to the aquatic food web.